# Regret Bounds without Lipschitz Continuity: Online Learning with Relative-Lipschitz Losses

**Yihan Zhou**\*
University of British Columbia

**Victor S. Portella**\*
University of British Columbia

**Mark Schmidt**
University of British Columbia
CCAI Affiliate Chair (Amii)

**Nicholas J. A. Harvey**
University of British Columbia

## Abstract

In online convex optimization (OCO), Lipschitz continuity of the functions is commonly assumed in order to obtain sublinear regret. Moreover, many algorithms have only logarithmic regret when these functions are also strongly convex. Recently, researchers from convex optimization proposed the notions of "relative Lipschitz continuity" and "relative strong convexity". Both of the notions are generalizations of their classical counterparts. It has been shown that subgradient methods in the relative setting have performance analogous to their performance in the classical setting.

In this work, we consider OCO for relative Lipschitz and relative strongly convex functions. We extend the known regret bounds for classical OCO algorithms to the relative setting. Specifically, we show regret bounds for the follow the regularized leader algorithms and a variant of online mirror descent. Due to the generality of these methods, these results yield regret bounds for a wide variety of OCO algorithms. Furthermore, we further extend the results to algorithms with extra regularization such as regularized dual averaging.

## 1   Introduction

In online convex optimization (OCO), at each of many rounds a player has to pick a point from a convex set while an adversary chooses a convex function that penalizes the player's choice. More precisely, in each round $t \in \mathbb{N}$, the player picks a point $x_t$ from a fixed convex set $\mathcal{X} \subseteq \mathbb{R}^n$ and an adversary picks a convex function $f_t$ depending on $x_t$. At the end of the round, the player suffers a loss of $f_t(x_t)$. Besides modeling a wide range of online learning problems [Shalev-Shwartz, 2011], algorithms for OCO are often used in batch optimization problems due to their low computational cost per iteration. For example, the widely used stochastic gradient descent (SGD) algorithm can be viewed as a special case of online gradient descent [Hazan, 2016, Chapter 3] and AdaGrad [Duchi et al., 2011] is a foundational adaptive gradient descent method originally proposed in the OCO setting. The performance measure usually used for OCO algorithms is the *regret*. It is the difference between the cost incurred to the player and a comparison point $z \in \mathcal{X} \subseteq \mathbb{R}^n$ (usually with minimum cumulative loss), that is to say,

$$\text{Regret}_T(z) := \sum_{t=1}^{T} f_t(x_t) - \sum_{t=1}^{T} f_t(z).$$

---

Classical results show that if the cost functions are Lipschitz continuous, then there are algorithms which suffer at most $O(\sqrt{T})$ regret in $T$ rounds [Zinkevich, 2003]. Additionally, if the cost functions are strongly convex, there are algorithms that suffer at most $O(\log T)$ regret in $T$ rounds [Hazan et al., 2007b]). However, not all loss functions that appear in applications, such as in inverse Poisson problems [Antonakopoulos et al., 2020] and support vector machines training [Lu, 2019], satisfy these conditions on the entire feasible set.

Recently, there has been a line of work investigating the performance of optimization methods beyond conventional assumptions [Bauschke et al., 2017, Lu et al., 2018, Lu, 2019]. Intriguingly, much of this line of work proposes relaxed assumptions under which classical algorithms enjoy convergence rates similar to the ones from the classical setting.

In particular, Lu [2019] proposed the notion of relative Lipschitz-continuity and showed how mirror descent (with properly chosen regularizer/mirror map) converges at a rate of $O(1/\sqrt{T})$ in $T$ iterations for non-smooth relative Lipschitz-continuous functions. Furthermore, they show a $O(1/T)$ convergence rate when the function is also relatively strongly-convex (a notion proposed by Lu et al. [2018]). Although the former result can be translated to a $O(\sqrt{T})$ regret bound for *online mirror descent* (OMD), the latter does not directly yield regret bounds in the online setting. Moreover, Orabona and Pál [2018] showed that OMD is not suitable when we do not know a priori the number of iterations since it may suffer linear regret in this case. Finally, at present it is not known how foundational OCO algorithms such as *follow the regularized leader* (FTRL) [Shalev-Shwartz, 2011, Hazan, 2016] and *regularized dual averaging* [Xiao, 2010] (RDA) perform in the relative setting.

**Our results.** We analyze the performance of two general OCO algorithms: FTRL and dual-stabilized OMD (DS-OMD, see [Fang et al., 2020]). We give $O(\sqrt{T})$ regret bounds in $T$ rounds for relative Lipschitz loss functions. Moreover, this is the first paper to show $O(\log T)$ regret if the loss functions are also relative strongly-convex.[1] In addition, we are able to extend these bounds for problems with composite loss functions, such as adding the $\ell_1$-norm to induce sparsity. The generality of these algorithms lead to regret bounds for a wide variety of OCO algorithms (see Shalev-Shwartz [2011], Hazan [2016] for some reductions). We demonstrate this flexibility by deriving convergence rates for *dual averaging* Nesterov [2009] and *regularized dual averaging* [Xiao, 2010].

## 1.1 Related Work

Analyses of gradient descent methods in the differentiable convex setting usually require the objective function $f$ to be Lipschitz smooth, that is, the gradient of the objective function $f$ is Lipschitz continuous. Bauschke et al. [2017] proposed a generalized Lipschitz smoothness condition, called *relative Lipschitz smoothness*, using Bregman divergences of a fixed reference function. They proposed a proximal mirror descent method[2] called NoLips with a $O(1/T)$ convergence rate for such functions. Van Nguyen [2017] independently developed similar ideas for analyzing the convergence of a Bregman proximal gradient method applied to convex composite functions in Banach spaces. Bolte et al. [2018] extended the framework of Bauschke et al. [2017] to the non-convex setting. Building upon this work, Lu et al. [2018] slightly relaxed the definition of relative smoothness and gave simpler analyses for mirror descent and dual averaging. Hanzely and Richtárik [2018] propose and analyse coordinate and stochastic gradient descent methods for relatively smooth functions. These ideas were later applied to non-convex problems by Mukkamala and Ochs [2019]. More recently, Gao et al. [2020] analysed the coordinate descent method with composite Lipschitz smooth objectives. Unlike those prior works, in this paper we focus on the online case with non-differentiable loss functions.

For non-differentiable convex optimization, Lipschitz continuity of the objective function is usually needed to obtain a $O(1/\sqrt{T})$ convergence guarantee for classical methods. Lu [2019] showed that this condition can be relaxed to what they called relative Lipschitz continuity of the objective function. Under this latter assumption, they gave $O(1/\sqrt{T})$ convergence rates for deterministic and stochastic

mirror descent. In a similar vein, Grimmer [2019] showed how projected subgradient descent enjoys a $O(1/\sqrt{T})$ convergence rate without Lipschitz continuity given that one has some control on the norm of the subgradients. None of these works considered online algorithms. Although the results from Lu [2019] for mirror descent can be adapted to the online setting, it is not clear how other foundational OCO algorithms such as FTRL or RDA perform in this setting.

Antonakopoulos et al. [2020] generalized the Lipschitz continuity condition from the perspective of Riemannian geometry. They proposed the notion of Riemann-Lipschitz continuity (RLC) and analyzed how OCO algorithms perform in this setting. They showed $O(\sqrt{T})$ regret bounds for both FTRL and OMD with RLC cost functions in both the online and stochastic settings. In Appendix A we discuss in detail the relationship between RLC and relative Lipschitzness and how some of our regret bounds compare to those due to Antonakopoulos et al. [2020]. In related work, Maddison et al. [2018] relaxed the Lipschitz smoothness condition by proposing a new family of optimization methods motivated from physics, to be more specific, the conformal Hamiltonian dynamics.

Moreover, in the presence of both Lipschitz continuity and strong convexity we can obtain $O(1/T)$ convergence rates in classical convex optimization [Bubeck, 2015, Section 3.4.1] and $O(\log T)$ regret in the online case [Hazan et al., 2007b]. By replacing the squared norm in the usual strong convexity inequality by a Bregman divergence of a fixed reference function yields the notion of *relative strong convexity*. This idea dates back to the work of Hazan et al. [2007a]. In recent work, Lu et al. [2018] showed algorithms with $O(1/T)$ convergence rates in the offline setting when the objective function is both relative Lipschitz continuous and relative strongly convex. Still, this latter work does not obtain regret bounds for the online case. Hazan et al. [2007a] analyze the online case and show logarithmic regret bounds for online mirror descent when the cost functions are strongly convex relative to the mirror map. However, they assume (classical) strong convexity of the mirror map, which ultimately implies that the cost function need also be strongly convex.[3] To the best of our knowledge, this is the first work studying conditions beyond strong convexity (and exp-concavity [Hazan et al., 2007b]) to obtain logarithmic regret bounds.

## 2 Formal Definitions

Throughout this paper, $\mathbb{R}^n$ denotes a $n$-dimensional real vector space endowed with an inner-product $\langle \cdot, \cdot \rangle$ and norm $\|\cdot\|$. We take $\mathcal{X} \subseteq \mathbb{R}^n$ to be a fixed convex set. The **dual norm** of $\|\cdot\|$ is defined by $\|x\|_* := \sup_{y \in \mathbb{R}^n \, : \, \|y\| \le 1} \langle x, y \rangle$ for each $x \in \mathbb{R}^n$. Moreover, for any convex function $f \colon \mathcal{X} \to \mathbb{R}$ and any $x \in \mathbb{R}^n$, a vector $g \in \mathbb{R}^n$ is a **subgradient** of $f$ at $x$ if $G$ satisfies the *subgradient inequality*

$$f(z) \ge f(x) + \langle g, x - z \rangle, \qquad \forall z \in \mathbb{R}^n. \tag{2.1}$$

We denote by $\partial f(x)$ the set of all subgradients of $f$ at $x$, called the **subdifferential** of $f$ at $x$. The **normal cone** of $\mathcal{X}$ at a point $x \in \mathcal{X}$ is the set $N_{\mathcal{X}}(x) := \{\, a \in \mathbb{R}^n : \langle a, z - x \rangle \le 0 \text{ for all } z \in \mathcal{X} \}$.

Let $R \colon \mathcal{D} \to \mathbb{R}$ be a convex function such that it is differentiable in $\mathcal{D}^\mathrm{o} := \mathrm{int}\, \mathcal{D}$ and such that we have $\mathcal{X} \subseteq \mathcal{D}^\mathrm{o}$. The **Bregman divergence** (with respect to $R$) is given by

$$D_R(x, y) := R(x) - R(y) - \langle \nabla R(y), x - y \rangle, \qquad \forall x \in \mathcal{D}, y \in \mathcal{D}^\mathrm{o}.$$

An interesting and useful identity regarding Bregman divergences, sometimes called *three-point identity* [Bubeck, 2015], is

$$D_R(x, y) + D_R(z, x) - D_R(z, y) = \langle \nabla R(x) - \nabla R(y), x - z \rangle, \qquad \forall z \in \mathcal{D}, \forall x, y \in \mathcal{D}^\mathrm{o}. \tag{2.2}$$

Although the Bregman divergence with respect to $R$ is not a metric, we can still interpret $D_R$ as a way of measuring distances through the lens of $R$. An instructive example is the Bregman divergence associated with the squared $\ell_2$-norm $R := \frac{1}{2}\|\cdot\|_2^2$. In this case, we have $D_R(x, y) = \frac{1}{2}\|x - y\|_2^2$ for all $x, y \in \mathbb{R}^n$, that is, the divergence boils down to the squared $\ell_2$-distance. In light of this, a possible way to generalize Lipschitz continuity and strong convexity is to replace the norm in the classical definitions by the square root of the Bregman divergence [Lu et al., 2018].

First, recall that a function $f\colon \mathcal{X} \to \mathbb{R}$ is $L$-**Lipschitz continuous** with respect to $\|\cdot\|$ on $\mathcal{X}' \subseteq \mathcal{X}$ if
$$|f(x) - f(y)| \leq L\|x - y\|, \qquad \forall x, y \in \mathcal{X}'.$$
Additionally, if $f$ is convex, then the above definition implies[4] that $\|g\|_* \leq L$ for all $x \in \mathcal{X}$ and all $g \in \partial f(x)$. Recall as well that a convex function $f\colon \mathcal{X} \to \mathbb{R}$ is $M$-**strongly convex** with respect to $\|\cdot\|$ on $\mathcal{X}' \subseteq \mathcal{X}$ for some $M > 0$ if
$$f(y) \geq f(x) + \langle g, y - x \rangle + \frac{M}{2}\|y - x\|^2, \qquad \forall x, y \in \mathcal{X}', \forall g \in \partial f(x).$$
Let us now state generalizations of the above definitions due to Lu et al. [2018] and Lu [2019].

**Definition 2.1** (Relative Lipschitz continuity)**.** A convex function $f\colon \mathcal{X} \to \mathbb{R}$ is $L$-**Lipschitz continuous** relative to $R$ if
$$\langle g, x - y \rangle \leq L\sqrt{2D_R(y, x)}, \qquad \forall x, y \in \mathcal{X}, \forall g \in \partial f(x).$$

In particular, if $f\colon \mathcal{X} \to \mathbb{R}$ is $L$-Lipschitz continuous relative to $R$, then
$$f(x) - f(y) \overset{(2.1)}{\leq} \langle g, x - y \rangle \leq L\sqrt{2D_R(y, x)}, \qquad \forall x, y \in \mathcal{X}, \forall g \in \partial f(x). \tag{2.3}$$

The original definition of Lu [2019] requires $\|g\|_*\|x - y\| \leq L\sqrt{2D_R(x, y)}$ for all $x, y \in \mathcal{X}$ and $g \in \partial f(x)$. Since $\langle a, b \rangle \leq \|a\|_*\|b\|$ for any $a, b \in \mathbb{R}^n$, the above definition is slightly more general and does not depend on the choice of a norm.

**Definition 2.2** (Relative strong convexity [Lu et al., 2018])**.** A convex function $f\colon \mathcal{X} \to \mathbb{R}$ is $M$-**strongly convex** relative to $R$ if
$$f(y) \geq f(x) + \langle g, y - x \rangle + MD_R(y, x), \qquad \forall y, x \in \mathcal{X}, \forall g \in \partial f(x). \tag{2.4}$$

A notable special case of relative Lipschitz-continuity or relative strong convexity is when we pick $R := \frac{1}{2}\|\cdot\|_2^2$ and the classical definitions with respect to the $\ell_2$-norm are recovered.

**Example** (A function that is relative Lipschitz but not Lipschitz)**.** Consider the function $f$ given by $f(x) := x^2$ for each $x \in \mathbb{R}$. Since the derivative of $f$ is unbounded on $\mathbb{R}$, it is not Lipschitz continuous on the entire line. Define the function $R$ by $R(x) := 2x^4$ for all $x \in \mathbb{R}$. Then,
$$D_R(y, x) = 2y^4 - 2x^4 - 8x^3(y - x) = \frac{1}{2}(x^2 - y^2)^2 + x^2(x - y)^2 \geq x^2(x - y)^2, \qquad \forall x, y \in \mathbb{R}.$$
Thus, $(f'(x)(x - y))^2 = 4x^2(x - y)^2 \leq 2 \cdot 2D_R(y, x)$ for any $x, y \in \mathbb{R}^n$. That is, $f$ is $\sqrt{2}$-Lipschitz continuous relative to $R$.

Lu [2019] discusses more substantial examples in detail, such as training of support vector machines, and finding a point in the intersection of several ellipsoids. Furthermore, he also gives a systematic way of picking a reference function for any objective functions whose subgradients at $x$ have $\ell_2$-norm bounded by a polynomial in $\|x\|_2$. This useful construction allows many optimization problems to benefit from algorithms that are designed for the relative setting.

## 2.1 Conventions and Assumptions used Throughout the Paper

We collect here some additional notation and assumptions used throughout the paper.[5] First, $\mathcal{X} \subseteq \mathbb{R}^n$ denotes a closed convex set and $\{f_t\}_{t \geq 1}$ denotes a sequence of convex functions such that $f_t\colon \mathcal{X} \to \mathbb{R}$ is subdifferentiable[6] on $\mathcal{X}$ for each $t \geq 1$. We denote by $\{\eta_t\}_{t \geq 0}$ a sequence of scalars such that $\eta_t \geq \eta_{t+1} > 0$ for each $t \geq 0$. Moreover, $\mathcal{D} \subseteq \mathbb{R}^n$ denotes a convex set with non-empty interior $\mathcal{D}^o := \text{int}(\mathcal{D})$ such that $\mathcal{X} \subseteq \mathcal{D}^o$. This latter set will be the domain of the regularizer for FTRL and of the mirror map for OMD. Namely, in Section 3 we denote by $R\colon \mathcal{D} \to \mathbb{R}$ the *regularizer* of FTRL, a convex function which is differentiable on $\mathcal{D}^o$. In Section 5 we denote by $\Phi\colon \mathcal{D} \to \mathbb{R}$ the *mirror map* of online mirror descent (whose precise definition we postpone to Section 5).

# 3 Follow the Regularized Leader

The *follow the regularized leader* (FTRL) algorithm is a classical method for OCO. At each round, FTRL picks a point that minimizes the cost incurred by the previously seen functions plus a regularizer convex function (an *FTRL regularizer*). Intuitively, the latter helps the choices of the algorithm not to change too widely from one round to the next. In Algorithm 1 we formally outline the FTRL algorithm. It is well known [Hazan, 2016] that, in a game with $T$ rounds, FTRL with properly tuned step sizes suffers at most $O(\sqrt{T})$ regret against Lipschitz continuous functions.[7] When the loss functions are additionally strongly convex, FTRL suffers at most regret $O(\log T)$. In this section we describe one of our main results: the FTRL algorithm preserves these asymptotic regret guarantees in the relative setting.

---

**Algorithm 1** Follow the Regularized Leader (FTRL) Algorithm

Compute $x_1 \in \arg\min_{x \in \mathcal{X}} R(x)$
Set $F_0 := 0$
**for** $t = 1, 2, \dots$ **do**
    Observe $f_t$ and suffer cost $f_t(x_t)$
    Set $F_t := F_{t-1} + f_t = \sum_{i=1}^{t} f_i$
    Compute $x_{t+1} \in \arg\min_{x \in \mathcal{X}} \left( F_t(x) + \frac{1}{\eta_t} R(x) \right)$

---

The usual first step in the analyses of FTRL algorithms is to use basic properties of the iterates (without relying on convexity) to bound the algorithm's regret by easier-to-analyse terms. Such bounds are usually the sum of two terms: the "diameter" of the feasible set through the lens of the FTRL regularizer and a sum of the difference in "quality" between consecutive iterates. For a classic example, see [Shalev-Shwartz, 2011, Lemma 2.3]. For our analysis we shall use a slightly tighter bound given by the Strong FTRL Lemma due to McMahan [2017]. For the sake of completeness we give a proof of this lemma (and discuss its applications in the composite setting) in Appendix C.1.

**Lemma 3.1.** (Strong FTRL Lemma [McMahan, 2017]) Let $\{f_t\}_{t\geq 1}$ be a sequence of functions such that $f_t: \mathcal{X} \to \mathbb{R}$ for each $t \geq 1$. Let $\{\eta_t\}_{t\geq 1}$ be a positive non-increasing sequence. Let $R: \mathcal{X} \to \mathbb{R}$ be such that $\{x_t\}_{t\geq 1}$ given as in Algorithm 1 is properly defined. If $F_t: \mathcal{X} \to \mathbb{R}$ is defined as in Algorithm 1 for each $t \geq 1$, then,

$$\text{Regret}_T(z) \leq \sum_{t=0}^{T} \left( \frac{1}{\eta_t} - \frac{1}{\eta_{t-1}} \right) (R(z) - R(x_t)) + \sum_{t=1}^{T} \left( H_t(x_t) - H_t(x_{t+1}) \right) \qquad \forall T > 0,$$

where $\eta_0 := 1$, $\frac{1}{\eta_{-1}} := 0$, $x_0 := x_1$, and $H_t := F_t + \frac{1}{\eta_t} R$ for each $t \geq 1$.

## 3.1 Sublinear Regret with Relative Lipschitz Functions

In the following theorem we formally state our sublinear $O(\sqrt{T})$ regret bound of FTRL in $T$ rounds in the setting where the cost functions are Lipschitz continuous relative to the regularizer function used in the FTRL method. The proof, which we defer to Appendix C.2, boils down to bounding the terms $H_t(x_t) - H(x_{t+1})$ from the Strong FTRL Lemma by (roughly) $L^2 \eta_{t-1}/2$. We do so by combining the optimality conditions from the definition of the iterates in Algorithm 1 with the $L$-Lipschitz continuity relative to $R$ of the loss functions.

**Theorem 3.2.** Let $\{x_t\}_{t\geq 1}$ be defined as in Algorithm 1 and suppose $f_t$ is $L$-Lipschitz continuous relative to $R$ for all $t \geq 1$. Let $z \in \mathcal{X}$ and let $K \in \mathbb{R}$ be such that $K \geq R(z) - R(x_1)$. Then,

$$\text{Regret}_T(z) \leq \frac{K}{\eta_T} + \sum_{t=1}^{T} \frac{L^2 \eta_{t-1}}{2}, \qquad \forall T > 0.$$

In particular, if $\eta_t := \sqrt{K}/(L\sqrt{t+1})$ for each $t \geq 0$, then $\text{Regret}_T(z) \leq 2L\sqrt{K(T+1)}$.

## 3.2 Logarithmic Regret with Relative Strongly Convex Functions

Hazan et al. [2007b] showed that if the cost functions are not only Lipschitz continuous but strongly convex as well, then the *follow the leader* (FTL) method—FTRL without any regularizer—attains logarithmic regret. Similarly, in this section we show that if the cost functions are relative Lipschitz continuous and relative strongly convex, both relative to the same fixed function, then FTL suffers regret at most logarithmic in the number of rounds. The proof of the next theorem is similar to the proof of Theorem 3.2 and is deferred to Appendix C.3.

**Theorem 3.3.** Let $\{x_t\}_{t \geq 1}$ be defined as in Algorithm 1 with $R := 0$. Assume that $f_t$ is $L$-Lipschitz continuous and $M$-strongly convex relative to a differentiable convex function $h \colon \mathcal{D} \to \mathbb{R}$ for each $t \geq 1$. Then, for all $z \in \mathcal{X}$,

$$\mathrm{Regret}_T(z) \leq \frac{L^2}{2M}(\log(T) + 1), \qquad \forall T > 0.$$

One might wonder whether requiring both Lipschitz continuity and strong convexity relative to the same function is too restrictive. Indeed, let $f$ be both $L$-Lipschitz continuous and $M$-strongly convex relative to $R$. Moreover, assume $f$ is differentiable for the sake of simplicity. If $x^* \in \mathcal{X}$ is a minimizer of $f$ over $\mathcal{X}$, then optimality conditions imply $-\nabla f(x^*) \in N_{\mathcal{X}}(x^*)$. Thus, by the definition of relative strong convexity,

$$f(y) - f(x^*) \geq \langle \nabla f(x^*), y - x^* \rangle + M D_R(y, x^*) \geq M D_R(y, x^*), \qquad \forall y \in \mathcal{X}$$

At the same time, by relative Lipschitz continuity (see (2.3)) we have

$$f(y) - f(x^*) \leq L\sqrt{2D_R(x^*, y)}, \qquad \forall y \in \mathcal{X}.$$

This means that $f$ has a $\Omega(D_R(\cdot, x^*))$ lower-bound and a $O(\sqrt{D_R(x^*, \cdot)})$ upper-bound. If the Bregman divergence between $y$ and $x^*$ were to go to infinity as $y$ ranges over $\mathcal{X}$, for example when $\mathcal{X} = \mathbb{R}^n$ and $R$ is the squared $\ell_2$ norm, then the lower-bound would eventually exceed the upper-bound on $\mathcal{X}$. Therefore, relative Lipschitz continuity and relative strong convexity can only coexist when $\mathcal{X}$ and the Bregman divergence with respect to $R$ of a minimizer and any point in $\mathcal{X}$ are both bounded. Although this is a somewhat restrictive condition, classical logarithmic regret results such as the ones due to Hazan et al. [2007b] also only hold over bounded sets. Moreover, as the next example shows, there are cases where logarithmic regret is attainable but *do not* fit into classical logarithmic regret results.

**Example** (A class functions that are both relative Lipschitz continuous and relative strongly convex). Define $f := \frac{1}{p}\|\cdot\|_2^p$ for some $p \geq 2$ and suppose $\mathcal{X} = [-\alpha, \alpha]^n$ for some $\alpha > 0$. First, note that $\nabla f(x) = \|x\|_2^{p-2}x$ and $\nabla^2 f(x) = \|x\|_2^{p-2}I + (p-2)\|x\|_2^{p-4}xx^T$ for any $x \in \mathbb{R}^n$. By Proposition 5.1 in Lu [2019], $f$ is 1-continuous relative to $R := \frac{1}{2p}\|\cdot\|_2^{2p}$ on $\mathbb{R}^n$ since $\|\nabla f(x)\|_2^2 = \|x\|_2^{2(p-2)} \cdot \|x\|_2^2 = \|x\|_2^{2p-2}$. Moreover, to show that $f$ is $M$-strongly convex relative to $R$, it suffices to show that $f - MR$ is convex (see Proposition 1.1 in Lu et al. [2018]). For any $M > 0$ and $x \in \mathbb{R}^n$ we have

$$\nabla^2 f(x) - M\nabla^2 R(x) = \|x\|_2^{p-2}I + (p-2)\|x\|_2^{p-4}xx^T - M(\|x\|_2^{2p-2}I + (2p-2)\|x\|_2^{2p-4}xx^T),$$
$$= \|x\|_2^{p-2}(1 - M\|x\|_2^p)I + \|x\|_2^{p-4}(p - 2 - M(2p-2)\|x\|_2^p)xx^T,$$
$$\succeq \|x\|_2^{p-4}(1 - M\|x\|_2^p + p - 2 - M(2p-2)\|x\|_2^p)xx^T,$$
$$= \|x\|_2^{p-4}(p - 1 - M(2p-1)\|x\|_2^p)xx^T,$$

where the only inequality follows since $\|x\|_2^2 I \succeq xx^T$ for any $x \in \mathbb{R}^n$. By setting $M := \frac{p-1}{(2p-1)(\sqrt{n}\alpha)^p}$ we have

$$p - 1 - M(2p-1)\|x\|_2^p \geq p - 1 - M(2p-1)(\sqrt{n}\alpha)^p = 0, \qquad \forall x \in \mathcal{X} = [-\alpha, \alpha]^n.$$

Thus, we have $\nabla^2 f(x) - M\nabla^2 R(x) \succeq 0$. Therefore, $f - MR$ is convex, which implies that $f$ is strongly convex relative to $R$ on $\mathcal{X}$. Note that $f$ is not classically strongly convex (that is, strongly convex with respect to the $\ell_2$ norm) for $p \geq 4$. To see this, note that $\nabla^2 f(x) - MI$ is not positive semidefinite around $0$ for any $M > 0$, and thus $f - M\|\cdot\|_2^2$ is not convex around $0$ no matter how small we pick $M > 0$ to be.

# 4   Dual Averaging and Composite Loss Functions

FTRL is a cornerstone algorithm in OCO, but sometimes it is not practical. Each iterate requires *exact* minimization of the loss functions (plus the regularizer) which might not have always a closed form solution. A notable special case of FTRL that mitigates this problem is the (online) *dual averaging* (DA) method whose offline version is due to Nesterov [2009]. In each iteration, DA picks a point from $\mathcal{X}$ that minimizes the sum of past subgradients (scaled by the step size) plus a FTRL regularizer $R$. Formally, for real convex functions $\{f_t\}_{t \geq 1}$ on $\mathcal{X}$, the online DA method computes iterates $\{x_t\}_{t \geq 1}$ such that

$$x_{t+1} \in \operatorname*{arg\,min}_{x \in \mathcal{X}} \Big( \eta_t \sum_{i=1}^{t} \langle g_i, x \rangle + R(x) \Big) \qquad \forall t \geq 0, \tag{4.1}$$

where $g_t \in \partial f_t(x_t)$ for each $t \geq 1$.

**Intuition.**   It is well-known that the DA algorithm reduces to FTRL applied to the linearized functions $\{\tilde{f}_t\}_{t \geq 1}$ given by $\tilde{f}_t := \langle g_t, \cdot \rangle$ for each $t \in \mathbb{N}$ (for details see Hazan [2016, Lemma 5.4]). This reduction obviously preserves the property of being Lipschitz continuous since the gradient of $\tilde{f}_t$ is $g_t$ everywhere. A natural idea would be to use this same reduction in the relative setting. Unfortunately, this reduction does not preserve the property of being relative Lipschitz! Luckily, our proof only requires a weaker condition: being "relative Lipschitz" at the particular point $x_t$. Namely, the relative $L$-Lipschitzness (see (2.3)) of $f_t$ implies $\langle \nabla \tilde{f}_t(x_t), x_t - y \rangle = \langle g_t, x_t - y \rangle \leq L\sqrt{2D_R(y, x_t)}$ for all $y \in \mathcal{X}$. That is all we need for the proof of Theorem 3.2 to go through, although we did state the theorem with this exact condition for the sake of simplicity. This discussion leads to the following corollary of Theorem 3.2.

**Corollary 4.1.** Let $\{x_t\}_{t \geq 1}$ be defined as in (4.1) and suppose $f_t$ is $L$-Lipschitz continuous relative to $R$ for all $t \geq 1$. Let $z \in \mathcal{X}$ and let $K \in \mathbb{R}$ be such that $K \geq R(z) - R(x_1)$. If $\eta_t := \sqrt{2K}/(L\sqrt{t+1})$ for all $t \geq 1$, then $\operatorname{Regret}_T(z) \leq 2L\sqrt{K(T+1)}$.

Another important consideration for applications is a variant of OCO in which the loss functions are composite [Duchi et al., 2010, Xiao, 2010]. More specifically, in this case we have a known "extra regularizer" $\Psi$, a (not necessarily differentiable) convex function, and add it to the loss functions. The goal is to induce some kind of structure in the iterates, such as adding $\ell_1$-regularization to promote sparsity. Note that OCO algorithms would still apply in this setting by replacing the loss functions $f_t$ with $f_t + \Psi$ at each round $t$. However, in this case we are not exploiting the fact that the function $\Psi$ is *known*. In the case of the relative setting, for example, it may be the case that the loss functions $f_t$ are relative Lipschitz-continuous with respect to a certain function $R$, while $\Psi$ is not. In Appendix D we extend the sublinear (composite) regret bound of Theorem 3.2 and show how this yields convergence bounds for regularized dual averaging [Xiao, 2010] in the relative setting.

# 5   Dual-Stabilized Online Mirror Descent

The mirror descent algorithm is a generalization of the classical gradient descent method that was first proposed by Nemirovsky and Yudin [1983]. A modern treatment was first given by Beck and Teboulle [2003]. The algorithm fits almost seamlessly into the OCO setting via a variant known as online mirror descent (OMD) (see [Hazan, 2016]). Recently, Orabona and Pál [2018] showed that OMD with a dynamic learning rate may suffer *linear* regret. (A dynamic learning rate is useful when we do not known the number of iterations ahead of time.) Moreover, this can happen even in simple and well-studied scenarios such as in the problem of prediction with expert advice, which corresponds to OMD equipped with negative entropy as a mirror map. In general, they showed that this may happen in cases where the Bregman divergence (with respect to the mirror map chosen) *is not* bounded over the entire feasible set. To resolve this issue, Fang et al. [2020] proposed a modified version of OMD called *dual-stabilized online mirror descent* (DS-OMD). In contrast to classical OMD, the regret bounds for the dual-stabilized version depend only on the Bregman divergence between the feasible set and the *initial iterate*.

We formally describe the DS-OMD method in Algorithm 2. Compared to OMD, DS-OMD adds an extra step in the dual space to mix the current dual iterate with the dual of the initial point. This step at iteration $t$ is controlled by a stabilization parameter $\gamma_t$ and it can be seen as a way to "stabilize" the

algorithm in the dual space. Throughout this section we closely follow the notation and assumptions of Bubeck [2015, Chapter 4]. We assume that we have a **mirror map** for $\mathcal{X}$, that is, a differentiable strictly-convex function $\Phi\colon \mathcal{D} \to \mathbb{R}$ for $\mathcal{X}$ such that the gradient of $\Phi$ diverges on the boundary of $\mathcal{D}$, that is, $\lim_{x\to\partial\mathcal{D}}\|\nabla\Phi(x)\|_2 = \infty$ where $\partial\mathcal{D} := \mathcal{D} \setminus \mathcal{D}^\circ$. These conditions on the mirror map guarantee that the algorithm is well-defined (for example, they guarantee the existence and uniqueness of the last step of Algorithm 2).

---

**Algorithm 2** Dual-Stabilized Online Mirror Descent

---

**Input:** Stabilization coefficient $\gamma_t$ and an initial iterate $x_1 \in \mathcal{X}$.
  **for** $t = 1, 2, \dots$ **do**
      Observe $f_t$ and suffer cost $f_t(x_t)$
      Compute $g_t \in \partial f_t(x_t)$
      $\hat{x}_t := \nabla\Phi(x_t)$
      $\hat{w}_{t+1} := \hat{x}_t - \eta_t g_t$
      $\hat{y}_{t+1} := \gamma_t \hat{w}_{t+1} + (1 - \gamma_t)\hat{x}_1$
      $y_{t+1} := \nabla\Phi^*(\hat{y}_{t+1})$
      Compute $x_{t+1} \in \arg\min_{x\in\mathcal{X}} D_\Phi(x, y_{t+1}) = \Phi(x) - \Phi(y_{t+1}) - \langle \nabla\Phi(y_{t+1}), x - y_{t+1}\rangle$

---

## 5.1 Sublinear Regret with Relative Lipschitz Functions

In this section, we give a regret bound for DS-OMD when the cost functions are all Lipschitz continuous relative to the mirror map $\Phi$. In this setting, if we set the stabilization coefficients to be $\gamma_t := \eta_{t+1}/\eta_t$ and step size $O(1/\sqrt{t})$, DS-OMD obtains sublinear regret. This is formally stated in the following theorem.

**Theorem 5.1.** Let $\{x_t\}_{t\geq 1}$ be defined as in Algorithm 2 with $\gamma_t := \eta_{t+1}/\eta_t$ for each $t \geq 1$. Assume that $f_t$ is $L$-Lipschitz continuous relative to $\Phi$ for all $t \geq 1$. Let $z \in \mathcal{X}$ and $K \in \mathbb{R}$ be such that $K \geq D_\Phi(z, x_1)$. Then,

$$\text{Regret}_T(z) \leq \frac{K}{\eta_{T+1}} + \sum_{t=1}^{T} \frac{\eta_t L^2}{2}, \qquad \forall T > 0.$$

In particular, if $\eta_t := \sqrt{K}/L\sqrt{t}$ for each $t \geq 1$, then $\text{Regret}_T(z) \leq 2L\sqrt{K(T+1)}$.

The proof is based on Theorem E.3, which gives an abstract regret upper bound for DS-OMD. Next we compute specific upper bounds of $D_\Phi(x_t, w_{t+1})$ for each $t \geq 1$ by relative Lipschitz continuity to make the abstract regret bound more specific. The whole proof of Theorem 5.1 is given in Appendix E.1.

If we set each $f_t$ to be a fixed function $f$ and take average of all iterates, then we get the following convergence rate for classical convex optimization as a corollary.

**Corollary 5.2.** Let $\Phi$ be a mirror map for $\mathcal{X}$ and let $f\colon \mathcal{X} \to \mathbb{R}$ be a convex $L$-Lipschitz-continuous function relative to $\Phi$. Let $\{x_t\}_{t\geq 1}$ be given as in Algorithm 2 with loss functions $f_t := f$, step sizes $\eta_t := \sqrt{K}/L\sqrt{t}$ for some $K \geq \sup_{z\in\mathcal{X}} D_\Phi(z, x_1)$, and stabilization parameter $\gamma_t := \eta_{t+1}/\eta_t$. If $x^* \in \mathcal{X}$ is a minimizer of $f$, then,

$$f\left(\frac{1}{T}\sum_{t=1}^{T} x_t\right) - f(x^*) \leq \frac{2L\sqrt{2K}}{\sqrt{T}}.$$

This recovers the same bound up to constant $4\sqrt{2}/3$ in Theorem 4.3 in Lu [2019], if we take $k = T-1$ and $t_i = \frac{\sqrt{K}}{\sqrt{T}L}$ for $i \geq 0$ therein.

## 5.2 Logarithmic Regret with Relative Strongly Convex Functions

In Section 3.2 we showed that FTRL suffers at most logarithmic regret when the loss functions are Lipschitz continuous and strongly convex, both relative to the same fixed reference function. Similarly, we show that OMD suffers at most logarithmic regret if we have Lipschitz continuity and strong convexity, both relative to the mirror map $\Phi$. Interestingly, in this case the dual-stabilization step can be skipped (that is, we can use $\gamma_t := 1$ for all $t$) and Algorithm 2 boils down to classic OMD.

**Theorem 5.3.** Let $\{x_t\}_{t \geq 1}$ be given as in Algorithm 2 with $\gamma_t := 1$ for all $t \geq 1$. Assume that $f_t$ is $L$-Lipschitz continuous and $M$-strongly convex relative to $\Phi$ for all $t \geq 1$. If $z \in \mathcal{X}$ and $\eta_t = \frac{1}{tM}$ for each $t \geq 1$, then,

$$\text{Regret}_T(z) \leq \frac{L^2}{2M}(\log T + 1), \qquad \forall T > 0.$$

The proof involves modifications of Theorem 5.1 and is deferred to Appendix E.2.

### 5.3 Sublinear Regret with Composite Loss Functions

We can extend our regret bounds to the setting with composite cost functions with minor modifications to Algorithm 2. The classical version OMD adapted to this setting is due to Duchi et al. [2010] and is known by composite objective mirror descent (COMID). They showed that COMID generalizes much prior work like forward-backward splitting and derived new results on efficient matrix optimization with Schatten $p$-norms based on this framework. Details of the modification needed on Algorithm E.3 in this setting together with regret bounds can be found in Appendix E.3.

## 6 Conclusions and Discussion

In this paper we showed regret bounds for both FTRL and stabilized OMD in the relative setting proposed by Lu [2019]. All the results hold in the *anytime setting* in which we do not know the number of rounds/iterations beforehand. Additionally, we gave logarithmic regret bounds for both algorithms when the functions are relatively strongly convex, analogous to the results known in the classical setting. Finally, we extend our results to the setting of composite cost functions, which is pervasive in practice. These results open up the possibility of a new range of applications for OCO algorithms and may allow for new analysis for known problems with better dependence on the instance's parameters.

At the moment there are at least two interesting directions for future research. The first would be to investigate the connections among the different notions of relative smoothness, Lipschitz continuity, and strong convexity in the literature. Another is to investigate systematic ways of choosing a regularizer/mirror map for any given optimization problem. The latter was already an interesting questions before notions of relative Lipschitz continuity and strong convexity were proposed, but these new ideas give more flexibility in the choice of a regularizer.

## 7 Statement of Broader Impact

In this paper we study the performance of online convex optimization algorithms when the functions are not necessarily Lipschitz continuous, a requirement in classical regret bounds. This opens up the range of applications for which we can use OCO with good guarantees and guides how such parameters such as regularizers/mirror maps and step sizes should be chosen. It is our hope that this aids practitioners to develop more efficient ways to optimize and train their current models. Furthermore, we hope theoreticians to be inspired to delve deep into the setting of non-smooth optimization beyond Lipschitz continuity. It not only opens up the range of applications, but sheds light onto the fundamental conditions on the cost functions and regularizers/mirror maps needed for OCO algorithms to have good guarantees. Due to the theoretical nature of this work, we do not see potentially bad societal or ethical impacts.

### Acknowledgments

We would like to thank the three anonymous reviewers and the meta-reviewer for engaging with our work. Moreover, we are thankful for their useful suggestions regarding the logarithmic regret results and the relationship of relative Lipschitz continuity and Riemann-Lipschitz continuity [Antonakopoulos et al., 2020]. We are also thankful to Wu Lin for useful discussions during the development of this work. Finally, we are grateful to Francesco Orabona for identifying in the work of Hazan et al. [2007a] some relationship with our results and one of the first uses of relative strong convexity.

### Funding Disclosure

This research was partially supported by NSERC Discovery Grants, Canada Research Chairs, the CIFAR Learning in Machines and Brains program, and the Canada CIFAR AI Chair Program.

## Footnotes

[1]This can be seen as analogous to the known logarithmic regret bounds when the loss functions are strongly convex [Hazan et al., 2007b].

[2]They propose an algorithm in the general case with composite functions, but when we set $f := 0$ in their algorithm it boils down to classical mirror descent. In this case the novelty comes from the convergence analysis at a $O(1/T)$ rate without the use of classical Lipschitz smoothness.

[3]More precisely, the regret bound in [Hazan et al., 2007a, Theorem 1] requires the cost functions $(g_t)_{t \in \mathbb{N}}$ to be strongly convex relative to the mirror map $f$. In turn, the result also requires $f$ to be strongly convex (in the classical sense) with respect to a fixed norm $\|\cdot\|$. This implies that the cost functions $(g_t)_{t \in \mathbb{N}}$ are strongly convex w.r.t. $\|\cdot\|$ as well.

[4]On the boundary of $\mathcal{X}$ this implication is not as strong: we can only guarantee the existence of one subgradient with small norm. For our purposes this will not be of fundamental importance. For a more precise statement see [Ben-Tal and Nemirovski, 2001, §5.3]

[5]The only exception is Lemma 3.1, which does not need convexity or differentiability of any of the functions.

[6]This is not too restrictive since convex functions are subdifferentiable on the relative interior of their domains [Rockafellar, 1997, Theorem 23.4].

[7]The big-O notation in this case hides constants that may depend on the dimension and other properties of the problem at hand. The best dependence on the Lipschitz constant and "distance to the comparison point" is usually achieved when the loss functions are Lipschitz continuous and the FTRL regularizer is strongly convex, both with respect to the same norm.

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
