[Supplementary Material]

# A  Relationship with Riemann-Lipschitz Continuity

Antonakopoulos et al. [2020] introduced the idea of *Riemann-Lipschitz* continuity (RLC). They show how FTRL and OMD can be used when the cost functions are all RLC in a way that guarantees $O(\sqrt{T})$ regret. In this section we shall discuss the relationship between these two generalizations of Lipschitz continuity. Ultimately, we will see that our results are at least as general but that further study into the relationship between these ideas is needed. We note that we will closely follow the notation of Antonakopoulos et al. [2020] and shall not discuss Riemannian metrics in full generality.

Let $G\colon \mathbb{R}^n \to \mathbb{R}^{n\times n}$ be such that $G(x)$ is a symmetric positive definite matrix for all $x \in \mathcal{X} \setminus \{0\}$ and $G(0)$ is symmetric positive semidefinite. Then the **Riemannian metric** (induced by G) is the collection of bilinear pairings $\{\, \langle \cdot, \cdot \rangle_x : x \in \mathcal{X} \,\}$ defined by

$$\langle y, z \rangle_x \coloneqq y^{\mathsf{T}} G(x) z, \qquad \forall x, y, z \in \mathcal{X}.$$

For conciseness, we shall denote the above metric induced by $G$ simply as the metric $G$. Moreover, the local norm induced by such the metric $G$ on $x \in \mathcal{X}$ is naturally given by

$$\|z\|_x \coloneqq \sqrt{\langle z, G(x)z \rangle}, \qquad \forall z \in \mathcal{X}.$$

Let us now give the definition of Riemann-Lispchitz continuity.

**Definition A.1.** Let $L > 0$. A function $f\colon \mathcal{X} \to \mathbb{R}$ is $L$-**Riemann-Lipschitz continuous** (RLC) relative to a Riemannian metric $G$ if

$$|f(y) - f(x)| \le L \cdot \operatorname{dist}_G(x, y) \qquad \forall x, y \in \mathcal{X},$$

where $\operatorname{dist}_G(x, y)$ is the Riemannian distance[*] between $x$ and $y$ induced by the Riemannian metric $G$.

The above definition is notably hard to work with. In the case of differentiable functions, RLC boils down to a much simpler and more intuitive condition.

**Proposition A.2** ([Antonakopoulos et al., 2020, Proposition 1]). Suppose that $f\colon \mathcal{X} \to \mathbb{R}$ is differentiable. Then $f$ is $L$-RLC if and only if

$$\|\operatorname{grad} f(x)\|_x \le L \qquad \text{for all } x \in \mathcal{X}, \tag{A.1}$$

where[†] $\operatorname{grad} f(x) \coloneqq G(x)^{-1} \nabla f(x)$ is the Riemannian gradient of $f$ at $x$ with respect to the metric $G$.

Finally, Antonakopoulos et al. [2020] use the notion of a *strong convexity* of a closed convex function $R\colon \mathcal{X} \to \mathbb{R}$ with respect to a metric $G$. For the sake of conciseness and simplicity, we shall use the equivalent condition given by Antonakopoulos et al. [2020, Lemma 1] and assume that $R$ is differentiable, but the arguments of this section hold even if $R$ is a closed convex function with a continuous selection of subgradients. More specifically, a differentiable convex function $R$ is $K$-**strongly convex** with respect to the metric $G$ for $K > 0$ if

$$\frac{K}{2}\|x - y\|_x^2 \le D_R(y, x), \qquad \forall x, y \in \mathcal{X}.$$

We are now in place to discuss the relationship between the notions of relative Lipchitz continuity and RLC. First, one should note that Proposition A.2 requires differentiability to hold. Since the regret bounds in Antonakopoulos et al. [2020] rely on (A.1), they also rely on the cost functions being differentiable. Since most $O(\sqrt{T})$ regret bounds in the online convex optimization literature (as well as the regret bounds in this text) *do not* rely on differentiability of the cost functions, it would be interesting to investigate if differentiability of the cost functions is in fact needed for the regret bounds of Antonakopoulos et al. [2020] to hold. In particular, in a way similar to classic Lipschitz continuity, it might be the case that (A.1) holds for at least one subgradient (after transformation by the metric $G$) at each point $x \in \mathcal{X}$ in the non-differentiable case.

---

[*]Equal contributions.

[*]We do not give here the full definition of a Riemannian metric as given by Antonakopoulos et al. [2020] since it will not be used in any of our discussions.

[†]Here we overlook the case when $x = 0$ (and, thus, when $G(x)$ is not necessarily invertible), for the sake of simplicity.

Assuming that the cost functions are indeed differentiable, we can show that relative Lipschitz continuity is at least as general as RLC. In the following proposition we show that if $f$ is a RLC function with respect to a metric $G$ and if we have a differentiable convex function $R$ which is strongly convex w.r.t. $G$ (which is used as a regularizer or a mirror map in FTRL and OMD), then $f$ is Lipschitz continuous relative to $R$.

**Proposition A.3.** Let $f \colon \mathcal{X} \to \mathbb{R}$ be a differentiable convex function and let $R \colon \mathcal{X} \to \mathbb{R}$ be a differentiable convex function such that $R$ is $K$-strongly convex with respect to the Riemannian metric $G$. If $f$ is $L$-RLC with respect to $G$, then $f$ is $L'$-Lipschitz continuous relative to $R$ where we set $L' := L\sqrt{K/2}$.

*Proof.* Let $x \in \mathcal{X}$. First, note that

$$\|\operatorname{grad} f(x)\|_x^2 = \operatorname{grad} f(x)^\mathsf{T} G(x) \operatorname{grad} f(x) = \nabla f(x)^\mathsf{T} G(x)^{-1} G(x) G(x)^{-1} \nabla f(x)$$
$$= \nabla f(x)^\mathsf{T} G(x)^{-1} \nabla f(x) = \|\nabla f(x)\|_{x,*}^2,$$

where $\|\cdot\|_{x,*}$ is the dual norm of $\|\cdot\|_x$. Therefore, for any $y \in \mathcal{X}$,

$$
\begin{aligned}
\nabla f(x)^\mathsf{T}(x-y) &\le \|\nabla f(x)\|_{x,*}\|x-y\|_x && \text{(by the definition of dual norm),}\\
&\le L\|x-y\|_x, && \text{(by RLC),}\\
&\le L\sqrt{\frac{K}{2} D_R(y,x)}, && \text{(by strong convexity of } R \text{ w.r.t. } G). \qquad \square
\end{aligned}
$$

The above proposition shows that Riemann-Lipschitz continuity (together with a strongly convex function with respect to the Riemannian metric) implies relative Lipschitz continuity. Thus, our regret bounds can be seen as generalizations of the regret bounds due to Antonakopoulos et al. [2020]. Moreover, the modularity of our proofs makes it easier to extend the results to the different settings (as demonstrated to the extension of some regret bounds to the composite setting as shown in Section 4, for example ).

Regarding the implication in the other direction, that is, whether relative Lipschitz continuity implies Riemannian Lipschitz continuity with respect to some metric $G$, it is not clear if it holds in general. The problem is that we do not know a systematic way of obtaining a metric $G$ given a function $f$ Lipschitz continuous relative to a function $R$ such that $f$ is RLC with respect to $G$ *and* $R$ is strongly convex with respect to $G$. Still, in some examples such a metric $G$ does seem to exist. It is not clear at the moment if both concepts of Lipschitz continuity are equivalent or not.

## B  Arithmetic Inequalities

**Lemma B.1.** Let $\{a_t\}_{t\ge 1}$ be a non-negative sequence with $a_1 > 0$. Then,

$$\sum_{t=1}^{T} \frac{a_t}{\sqrt{\sum_{i=1}^{t} a_i}} \le 2\sqrt{\sum_{t=1}^{T} a_t}, \qquad \forall T \in \mathbb{N}.$$

*Proof.* The proof is by induction on $T$. The statement holds trivially for $T = 1$. Let $T > 1$ and define $s := \sum_{t=1}^{T} a_t$. By the induction hypothesis,

$$\sum_{t=1}^{T} \frac{a_t}{\sqrt{\sum_{i=1}^{t} a_i}} \le 2\sqrt{\sum_{t=1}^{T-1} a_t} + \frac{a_T}{\sqrt{\sum_{i=1}^{T} a_i}} = 2\sqrt{s - a_T} + \frac{a_T}{\sqrt{s}}.$$

Finally, note that

$$2\sqrt{s-a_T} + \frac{a_T}{\sqrt{s}} \le 2\sqrt{s} \iff 2\sqrt{s(s-a_T)} \le 2s - a_T \iff 4s(s-a_T) \le (2s-a_T)^2,$$
$$\iff 4s^2 - 4sa_T \le 4s^2 - 4sa_T + a_T^2 \iff 0 \le a_T^2. \qquad \square$$

# C Proofs for Section 3

## C.1 Strong FTRL Lemma

In this section we give a proof of Lemma 3.1 for completeness. We also show how the lemma can be used for the composite setting. For further discussions on the lemma and on FTRL, see the thorough survey of McMahan [2017].

*Proof of Lemma 3.1.* Fix $T > 0$. Define $r_t := (\frac{1}{\eta_t} - \frac{1}{\eta_{t-1}})R$ for each $t \geq 0$ (recall that $\eta_0 := 1$ and $1/\eta_{-1} := 0$), define $h_t := r_t + f_t$ for each $t \geq 1$, and set $h_0 := r_0$. In this way, we have

$$\sum_{i=0}^{t} h_t = \sum_{i=1}^{t} f_t + \sum_{i=0}^{t} r_t = \sum_{i=1}^{t} f_t + \frac{1}{\eta_t}R = H_t, \qquad \forall t \geq 0.$$

In particular,

$$x_t \in \arg\min_{x \in \mathcal{X}} H_{t-1}(x) = \arg\min_{x \in \mathcal{X}} \sum_{i=0}^{t-1} h_i(x), \qquad \forall t \geq 0. \tag{C.1}$$

Let us now bound the regret of the points $x_1, \ldots, x_T$ with respect to the functions $h_1, \ldots, h_T$ and to a comparison point $z \in \mathcal{X}$ (plus a $-h_0(z)$ term):

$$\sum_{t=1}^{T} (h_t(x_t) - h_t(z)) - h_0(z) = \sum_{t=1}^{T} h_t(x_t) - H_T(z) = \sum_{t=1}^{T} (H_t(x_t) - H_{t-1}(x_t)) - H_T(z),$$

$$\overset{(\text{C.1})}{\leq} \sum_{t=1}^{T} (H_t(x_t) - H_{t-1}(x_t)) - H_T(x_{T+1}),$$

$$= \sum_{t=1}^{T} (H_t(x_t) - H_t(x_{t+1})) - H_0(x_1),$$

where in the last equation we just re-indexed the summation, placing $H_{T+1}(x_{T+1})$ inside the summation, and leaving $H_0(x_1)$ out. Re-arranging the terms and using $H_0 = h_0 = r_0$ and $x_0 = x_1$ yield

$$\sum_{t=1}^{T} (f_t(x_t) + r_t(x_t) - f_t(z) - r_t(z)) = \sum_{t=1}^{T} (h_t(x_t) - h_t(z)),$$

$$\leq r_0(z) - r_0(x_0) + \sum_{t=1}^{T} (H_t(x_t) - H_t(x_{t+1})),$$

which implies

$$\text{Regret}_T(z) = \sum_{t=1}^{T} (f_t(x_t) - f_t(z)) \leq \sum_{t=0}^{T} (r_t(z) - r_t(x_t)) + \sum_{t=1}^{T} (H_t(x_t) - H_t(x_{t+1})).$$

Since $r_t = (\frac{1}{\eta_t} - \frac{1}{\eta_{t-1}})R$ for all $t \geq 0$, we have

$$\sum_{t=0}^{T} (r_t(z) - r_t(x_t)) = \sum_{t=0}^{T} \left(\frac{1}{\eta_t} - \frac{1}{\eta_{t-1}}\right)(R(z) - R(x_t)). \qquad \square$$

For the composite setting (see Section D), we modify the definition of $r_t$ for $t \geq 1$ (maintaining the definition of $r_0$) in the above proof for

$$r_t := \left(\frac{1}{\eta_t} - \frac{1}{\eta_{t-1}}\right)R + \Psi, \qquad \forall t \geq 1.$$

In this case, we have

$$H_t = \sum_{i=1}^{t} f_t + \sum_{i=0}^{t} r_t = \sum_{i=1}^{t} f_t + \frac{1}{\eta_t}R + t\Psi.$$

Proceeding in the same way as in the proof of Lemma 3.1, we get

$$\sum_{t=1}^{T}(f_t(x_t) - f(z)) \le \sum_{t=0}^{T}\left(\frac{1}{\eta_t} - \frac{1}{\eta_{t-1}}\right)(R(z) - R(x_t)),$$

$$+ \sum_{t=1}^{T}(\Psi(z) - \Psi(x_t)) + \sum_{t=1}^{T}(H_t(x_t) - H_t(x_{t+1})),$$

Re-arranging yields

$$\text{Regret}_T^{\Psi}(z) \le \sum_{t=0}^{T}\left(\frac{1}{\eta_t} - \frac{1}{\eta_{t-1}}\right)(R(z) - R(x_t)) + \sum_{t=1}^{T}(H_t(x_t) - H_t(x_{t+1})). \qquad \text{(C.2)}$$

## C.2 Sublinear Regret with Relative Lipschitz Functions

With the Strong FTRL Lemma, to derive regret bounds we can focus on bounding the difference in cost between consecutive iterates. In this section we will prove the sublinear regret bound for FTRL from Theorem 3.2. In the next lemma we give a bound on these costs based on the Bregman divergence of the FTRL regularizer, this time relying on convexity (but not on much more). Loosely saying, the first claim of the next lemma follows from the optimality conditions of the iterates of FTRL and the second follows from the subgradient inequality.

**Lemma C.1.** Let $\{x_t\}_{t\ge 1}$ and $\{F_t\}_{t\ge 0}$ be defined as in Algorithm 1. Then, for each $t \in \mathbb{N}$ there is $p_t \in N_{\mathcal{X}}(x_t)$ such that $-p_t - \frac{1}{\eta_{t-1}}\nabla R(x_t) \in \partial F_{t-1}(x_t)$, where $\eta_0 \in \mathbb{R}$ can be any positive constant. Moreover, this implies

$$F_{t-1}(x_t) - F_{t-1}(x_{t+1}) \le \frac{1}{\eta_{t-1}}\left(R(x_{t+1}) - R(x_t) - D_R(x_{t+1}, x_t)\right).$$

*Proof.* Let $t \ge 1$. By the definition of the FTRL algorithm, we have $x_t \in \arg\min_{x\in\mathcal{X}}(F_{t-1}(x) + \frac{1}{\eta_{t-1}}R(x))$. By the optimality conditions for convex programs, we have

$$\partial\left(F_{t-1} + \tfrac{1}{\eta_{t-1}}R\right)(x_t) \cap (-N_{\mathcal{X}}(x_t)) \ne \varnothing.$$

Since $\partial\left(F_{t-1} + \frac{1}{\eta_{t-1}}R\right)(x_t) = \partial F_{t-1}(x_t) + \frac{1}{\eta_{t-1}}\nabla R(x_t)$, the above shows there is $p_t \in N_{\mathcal{X}}(x_t)$ such that

$$-p_t - \frac{1}{\eta_{t-1}}\nabla R(x_t) \in \partial F_{t-1}(x_t).$$

Using the subgradient inequality (2.1) with the above subgradient yields,

$$\begin{aligned}
&F_{t-1}(x_t) - F_{t-1}(x_{t+1}) \\
&\le -\langle p_t, x_t - x_{t+1}\rangle - \tfrac{1}{\eta_{t-1}}\langle \nabla R(x_t), x_t - x_{t+1}\rangle, \\
&\le -\tfrac{1}{\eta_{t-1}}\langle \nabla R(x_t), x_t - x_{t+1}\rangle \qquad\qquad \text{(by the definition of normal cone),} \\
&= \tfrac{1}{\eta_{t-1}}\left(R(x_{t+1}) - R(x_t) - D_R(x_{t+1}, x_t)\right),
\end{aligned}$$

where in the last equation we used that, by definition of the Bregman divergence, $D_R(x_{t+1}, x_t) = R(x_{t+1}) - R(x_t) - \langle \nabla R(x_t), x_{t+1} - x_t\rangle$ and, thus, $-\langle \nabla R(x_t), x_t - x_{t+1}\rangle = R(x_{t+1}) - R(x_t) - D_R(x_{t+1}, x_t)$. $\qquad\square$

*Proof of Theorem 3.2.* For each $t \ge 0$ let $H_t$ be defined as in the Strong FTRL Lemma and fix $t \ge 0$. We have

$$H_t(x_t) - H_t(x_{t+1}) = F_t(x_t) - F_t(x_{t+1}) + \frac{1}{\eta_t}(R(x_t) - R(x_{t+1})). \qquad \text{(C.3)}$$

Using $F_t = F_{t-1} + f_t$ together with Lemma C.1 we have

$$F_t(x_t) - F_t(x_{t+1}) = F_{t-1}(x_t) - F_{t-1}(x_{t+1}) + f_t(x_t) - f_t(x_{t+1}),$$

$$\le \frac{1}{\eta_{t-1}}\left(R(x_{t+1}) - R(x_t) - D_R(x_{t+1}, x_t)\right) + f_t(x_t) - f_t(x_{t+1}).$$

Plugging the above inequality onto (C.3) yields

$$\text{(C.3)} \le f_t(x_t) - f_t(x_{t+1}) - \frac{D_R(x_{t+1}, x_t)}{\eta_{t-1}} + \Big(\frac{1}{\eta_t} - \frac{1}{\eta_{t-1}}\Big)(R(x_t) - R(x_{t+1})). \qquad \text{(C.4)}$$

Since $f_t$ is $L$-relative Lipschitz continuous with respect to $R$, we apply (2.3) followed by the the arithmetic-geometric mean inequality $\sqrt{\alpha\beta} \le (\alpha + \beta)/2$ with $\alpha := L^2\eta_{t-1}$ and $\beta := 2D_R(x_{t+1}, x_t)/\eta_{t-1}$ to get

$$f_t(x_t) - f_t(x_{t+1}) - \frac{D_R(x_{t+1}, x_t)}{\eta_{t-1}} \overset{(2.3)}{\le} L\sqrt{2D_R(x_{t+1}, x_t)} - \frac{D_R(x_{t+1}, x_t)}{\eta_{t-1}} \le \frac{L^2\eta_{t-1}}{2}.$$

Applying the above on (C.4) yields

$$\text{(C.4)} \le \frac{L^2\eta_{t-1}}{2} + \Big(\frac{1}{\eta_t} - \frac{1}{\eta_{t-1}}\Big)(R(x_t) - R(x_{t+1})).$$

Plugging the above inequality into the the Strong FTRL Lemma together with $R(x_1) \le R(x_t)$ for each $t \ge 1$ (which follows by the definition of $x_1$) yields

$$\begin{aligned}
\text{Regret}_T(z) &\le \sum_{t=0}^{T} \Big(\frac{1}{\eta_t} - \frac{1}{\eta_{t-1}}\Big)(R(z) - R(x_t) + R(x_t) - R(x_{t+1})) + \sum_{t=1}^{T} \frac{L^2\eta_{t-1}}{2}, \\
&= \sum_{t=0}^{T} \Big(\frac{1}{\eta_t} - \frac{1}{\eta_{t-1}}\Big)(R(z) - R(x_{t+1})) + \sum_{t=1}^{T} \frac{L^2\eta_{t-1}}{2}, \\
&\le \frac{1}{\eta_T}(R(z) - R(x_1)) + \sum_{t=1}^{T} \frac{L^2\eta_{t-1}}{2} \le \frac{K}{\eta_T} + \sum_{t=1}^{T} \frac{L^2\eta_{t-1}}{2}.
\end{aligned}$$

If we set $\eta_t := \sqrt{2K}/(L\sqrt{t+1})$ and since $\sum_{t=1}^{T} \frac{1}{\sqrt{t}} \le 2\sqrt{T}$ by Lemma B.1 in Appendix B, then

$$\text{Regret}_T(z) \le L\sqrt{K(T+1)} + \frac{L\sqrt{K}}{2} \sum_{t=1}^{T} \frac{1}{\sqrt{t}} \le L\sqrt{K(T+1)} + L\sqrt{KT} \le 2L\sqrt{K(T+1)}. \quad \square$$

## C.3 Logarithmic Regret

The next lemma strengthens the bound from Lemma C.1 in the case where the loss functions are relative strongly convex with respect to a fixed reference function. We further simplify matters by taking $R = 0$, that is, regularization is not needed for FTRL in the relative strongly convex case.

**Lemma C.2.** Let $\{x_t\}_{t\ge1}$ be defined as in Algorithm 1 with $R := 0$. Moreover, let $h\colon \mathcal{D} \to \mathbb{R}$ be a differentiable convex function such that $f_t$ is $M$-strongly convex relative to $h$ for each $t \ge 1$. Then, for all $T \ge 1$,

$$F_{t-1}(x_t) - F_{t-1}(x_{t+1}) \le -(t-1)MD_h(x_{t+1}, x_t).$$

*Proof.* Let $t \ge 1$. Note that $F_{t-1}$ is $(t-1)M$-strongly convex relative to $R$ since it is the sum of $t-1$ functions that are each $M$-strongly convex relative to $R$. Additionally, let $p_t \in N_{\mathcal{X}}(x_t)$ be as given by Lemma C.1. By this lemma we have $-p_t \in \partial F_{t-1}(x_t)$. Thus, using inequality (2.4) from the definition of relative strong convexity with this subgradient yields

$$F_{t-1}(x_t) - F_{t-1}(x_{t+1}) \le -\langle p_t, x_t - x_{t+1}\rangle - (t-1)MD_h(x_{t+1}, x_t).$$

By the definition of normal cone we have $-\langle p_t, x_t - x_{t+1}\rangle = \langle p_t, x_{t+1} - x_t\rangle \le 0$, which yields the desired inequality. $\square$

*Proof of Theorem 3.3.* For each $t \ge 0$ let $H_t\colon \mathcal{X} \to \mathbb{R}$ be defined as in the Strong FTRL Lemma and fix $t \ge 0$. Since $R = 0$, we have $H_t = F_t$. This together with Lemma C.2 yields

$$\begin{aligned}
H_t(x_t) - H_t(x_{t+1}) = F_t(x_t) - F_t(x_{t+1}) &= F_{t-1}(x_t) - F_{t-1}(x_{t+1}) + f_t(x_t) - f_t(x_{t+1}), \\
&\le -(t-1)MD_h(x_{t+1}, x_t) + f_t(x_t) - f_t(x_{t+1}). \qquad \text{(C.5)}
\end{aligned}$$

Let $g_t \in \partial f_t(x_t)$. Since $f_t$ is $L$-Lipschitz continuous and $M$-strongly convex, both relative to $h$, we have

$$f_t(x_t) - f_t(x_{t+1}) \overset{(2.4)}{\leq} \langle g_t, x_t - x_{t+1} \rangle - M D_h(x_{t+1}, x_t) \overset{(2.3)}{\leq} L\sqrt{2D_R(x_{t+1}, x_t)} - M D_R(x_{t+1}, x_t).$$

Applying the above to (C.5) together with the fact that $\sqrt{\alpha\beta} \leq (\alpha + \beta)/2$ with $\alpha := L^2/(Mt)$ and $\beta := 2tM D_R(x_{t+1}, x_t)$ yields

$$H_t(x_t) - H_t(x_{t+1}) \leq L\sqrt{2D_R(x_{t+1}, x_t)} - tM D_R(x_{t+1}, x_t) \leq \frac{L^2}{2Mt}.$$

Finally, plugging the above inequality into the Strong FTRL Lemma (with $R = 0$) gives

$$\text{Regret}_T(z) \leq \sum_{t=0}^{T}(H_t(x_t) - H_t(x_{t+1})) \leq \frac{L^2}{2M}\sum_{t=1}^{T}\frac{1}{t} \leq \frac{L^2}{2M}(\log(T) + 1). \qquad \square$$

## D   Sublinear Regret Bounds for FTRL with Composite Loss Functions

In this section we extend the results from Section 3 to the case where the loss functions are *composite*. Specifically, there is a known non-negative convex function $\Psi \colon \mathcal{X} \to \mathbb{R}_+$ (sometimes called *extra regularizer*) which is subdifferentiable on $\mathcal{X}$ and at round $t$ the loss function presented to the player is $f_t + \Psi$. Usually $\Psi$ is a simple function which is easy to optimize over (such as the $\ell_1$-norm). Thus, although $f_t + \Psi$ might not preserve relative Lipschitz continuity of $f_t$, one might still hope to obtain good regret bounds in this case. We shall see that FTRL does not need any modifications to enjoy of good theoretical guarantees in this setting. Yet, its analysis in the composite case will allow us to derive regret bounds for the *regularized dual averaging* method due to Xiao [2010].

In the composite case we measure the performance of an OCO algorithm by its **composite regret** (against a point $z \in \mathcal{X}$) given by

$$\text{Regret}_T^{\Psi}(z) := \sum_{t=1}^{T}(f_t(x_t) + \Psi(x_t)) - \inf_{z \in \mathcal{X}}\sum_{t=1}^{T}(f_t(z) + \Psi(z)), \qquad \forall T > 0. \tag{D.1}$$

In the case of FTRL, practically no modifications to the algorithm are needed. Namely, the update of Algorithm 1 becomes

$$x_{t+1} \in \arg\min_{x \in \mathcal{X}}\Big(\sum_{i=1}^{t} f_i(x) + t\Psi(x) + \frac{1}{\eta_t}R(x)\Big), \qquad \forall t \geq 0.$$

We do make the additional assumption that $\Psi(x_1) = 0$, that is, $x_1$ minimizes $\Psi$ and tha latter has minimum value of $0$. In practice one has some control on $\Psi$, so this assumption is not too restrictive. The next theorem shows that we can recover the regret bound from Theorem 3.2 for the composite setting even if $\Psi$ is not relative Lipschitz-continuous with respect to the FTRL regularizer.

**Theorem D.1.** Let $\Psi \colon \mathcal{X} \to \mathbb{R}_+$ be a nonnegative convex function such that $\{x_t\}_{t \geq 1}$ as given as in Algorithm 1 are such that $\Psi(x_1) = 0$. Assume that $f_t$ is $L$-Lipschitz continuous relative to $R$ for all $t \geq 1$. Let $z \in \mathcal{X}$ and $K \in \mathbb{R}$ be such that $K \geq R(z) - R(x_1)$. Additionally, assume $\Psi(x_1) = 0$. Then,

$$\text{Regret}_T^{\Psi}(z) \leq \frac{2K}{\eta_T} + \sum_{t=1}^{T}\frac{L^2\eta_{t-1}}{2}, \qquad \forall T > 0.$$

In particular, if $\eta_t := \sqrt{2K}/(L\sqrt{t+1})$ for each $t \geq 1$, then $\text{Regret}_T^{\Psi}(z) \leq 2L\sqrt{K(T+1)}$

The proof is largely identical to the proof of Theorem 3.2. One of the main differences in the analysis is the following version of Lemma C.1 tweaked for the composite setting. It follows by adding $(t-1)\Psi$ to $F_{t-1}$ in the proof of the original lemma and using the properties of the subgradient. We give the full proof for the sake of completeness.

**Lemma D.2.** Let $\Psi \colon \mathcal{X} \to \mathbb{R}_+$ be a nonnegative convex function such that $\{x_t\}_{t \geq 1}$ as given as in Algorithm 1 are such that $\Psi(x_1) = 0$. Then, for each $t \in \mathbb{N}$ there is $p_t \in N_{\mathcal{X}}(x_t)$ such that

$$-p_t - \frac{1}{\eta_{t-1}}\nabla R(x_t) \in \partial\big(F_{t-1} + (t-1)\Psi\big)(x_t),$$

and the above implies

$$F_{t-1}(x_t) - F_{t-1}(x_{t+1}) + (t-1)(\Psi(x_t) - \Psi(x_{t+1}))$$
$$\leq \frac{1}{\eta_{t-1}}\big(R(x_{t+1}) - R(x_t) - D_R(x_{t+1}, x_t)\big)(t-1).$$

*Proof.* Let $t \geq 1$. By the definition of the FTRL algorithm, we have $x_t \in \arg\min_{x\in\mathcal{X}}(F_{t-1}(x) + (t-1)\Psi(x) + \frac{1}{\eta_{t-1}}R(x))$. By the optimality conditions for convex programs, we have

$$\partial\Big(F_{t-1} + (t-1)\Psi(x) + \frac{1}{\eta_{t-1}}R\Big)(x_t) \cap (-N_{\mathcal{X}}(x_t)) \neq \varnothing.$$

Since $\partial(F_{t-1} + (t-1)\Psi(x) + \frac{1}{\eta_{t-1}}R)(x_t) = \partial(F_{t-1} + (t-1)\Psi(x))(x_t) + \frac{1}{\eta_{t-1}}\nabla R(x_t)$, the above shows there is $p_t \in N_{\mathcal{X}}(x_t)$ such that

$$-p_t - \frac{1}{\eta_{t-1}}\nabla R(x_t) \in \partial(F_{t-1} + (t-1)\Psi(x))(x_t).$$

Using the subgradient inequality (2.1) with the above subgradient yields,

$$F_{t-1}(x_t) + (t-1)\Psi(x_t) - F_{t-1}(x_{t+1}) - (t-1)\Psi(x_{t+1})$$
$$\leq -\langle p_t, x_t - x_{t+1}\rangle - \frac{1}{\eta_{t-1}}\langle \nabla R(x_t), x_t - x_{t+1}\rangle,$$
$$\leq -\frac{1}{\eta_{t-1}}\langle \nabla R(x_t), x_t - x_{t+1}\rangle \qquad\qquad \text{(by the definition of normal cone)},$$
$$= \frac{1}{\eta_{t-1}}\big(R(x_{t+1}) - R(x_t) - D_R(x_{t+1}, x_t)\big),$$

where in the last equation we used that, by definition of the Bregman divergence, $D_R(x_{t+1}, x_t) = R(x_{t+1}) - R(x_t) - \langle \nabla R(x_t), x_{t+1} - x_t\rangle$ and, thus, $-\langle\nabla R(x_t), x_t - x_{t+1}\rangle = R(x_{t+1}) - R(x_t) - D_R(x_{t+1}, x_t)$. □

Now we are in position to prove Theorem D.1.

*Proof of Theorem D.1.* We proceed in a way extremely similar to the proof of Theorem 3.2, but in place of the standard FTRL Lemma we use its composite version as in (C.2).

For each $t \geq 0$ let $H_t$ be define das in the (composite) Strong FTRL Lemma so that $H_t = \sum_{i=1}^t f_i + t\Psi + \frac{1}{\eta_t}R$ and fix $t \geq 0$. In this case we have

$$H_t(x_t) - H_t(x_{t+1}) = F_t(x_t) - F_t(x_{t+1}) + t(\Psi(x_t) - \Psi(x_{t+1})) + \frac{1}{\eta_t}(R(x_t) - R(x_{t+1})).$$

Using $F_t = F_{t-1} + f_t$ together with Lemma D.2 we have

$$F_t(x_t) - F_t(x_{t+1}) + t(\Psi(x_t) - \Psi(x_{t+1}))$$
$$\leq \frac{1}{\eta_{t-1}}\big(R(x_{t+1}) - R(x_t) - D_R(x_{t+1}, x_t)\big) + f_t(x_t) - f_t(x_{t+1}) + \Psi(x_t) - \Psi(x_{t+1}).$$

Proceeding as in the proof of Theorem 3.2 (with the addition of a $\Psi(x_t) - \Psi(x_{t+1})$ term) we have

$$H_t(x_t) - H_t(x_{t+1}) \leq \frac{L^2\eta_{t-1}}{2} + \Big(\frac{1}{\eta_t} - \frac{1}{\eta_{t-1}}\Big)(R(x_t) - R(x_{t+1})) + \Psi(x_t) - \Psi(x_{t+1}).$$

When summing over $t \in \{1, \dots, T\}$, the terms $\Psi(x_t) - \Psi(x_{t+1})$ telescope so that, since $x_1$ minimizes $\Psi$, we have

$$\sum_{t=1}^T (\Psi(x_t) - \Psi(x_{t+1}) = \Psi(x_1) - \Psi(x_{T+1}) \leq 0.$$

Therefore, the remainder of the proof follows as in the proof of Theorem 3.2. □

## D.1  Regularized Dual Averaging

As previously discussed, applying OCO algorithms such as dual averaging in an out-of-the-box fashion when the loss functions are composite case does not exploit the structure of the extra-regularization given by $\Psi$ and may have poor performance in practice. For example, McMahan [2017] shows that applying DA in the composite case with $\Psi := \|\cdot\|_1$ does not yield sparse solutions. Xiao [2010] proposed the *regularized dual averaging* (RDA) method to solve this issue. The algorithm is identical to DA but it *does not linearize* the function $\Psi$. Formally, the initial iterate $x_1$ is in $\arg\min_{x\in\mathcal{X}}(R(x)$ and is such that $\Psi(x_1) = 0$, that is, $x_1$ minimizes $\Psi$. For the following rounds, RDA computes

$$x_{t+1} \in \arg\min_{x\in\mathcal{X}}\Big(\sum_{i=1}^{t}\langle g_i, x\rangle + t\Psi(x) + \frac{1}{\eta_t}R(x)\Big) \qquad \forall t \geq 1. \tag{D.2}$$

With an argument analogous to the one made in Section 4, we can write RDA as an instance of FTRL (with composite loss functions) and obtain the following corollary of Theorem D.1.

**Corollary D.3.** Let $\Psi\colon \mathbb{R}^n \to \mathbb{R}_+$ be a nonnegative convex function. Let $\{x_t\}_{t\geq 1}$ be defined as in (D.2) and assume $\Psi(x_1) = 0$. Moreover, suppose $f_t$ is $L$-Lipschitz continuous relative to $R$ for all $t \geq 1$. Let $z \in \mathcal{X}$ and let $K \in \mathbb{R}$ be such that $K \geq R(z) - R(x_1)$. If $\eta_t := \sqrt{2K}/(L\sqrt{t+1})$ for all $t \geq 1$, then $\mathrm{Regret}_T^{\Psi}(z) \leq 2L\sqrt{K(T+1)}$.

# E  Proofs for Section 5

In this section we give the missing proofs of Section 5. Throughout this section, let $\{x_t\}_{t\geq 1}$ and $\{\hat{w}_t\}_{t\geq 1}$ be defined as in Algorithm 2, and define

$$w_t := \nabla\Phi^*(\hat{w}_t), \qquad \forall t \geq 1.$$

First, let us state inequality (4.9) and Claim 4.2 (without substituting exactly value of $\gamma_t$) from Fang et al. [2020] at the beginning, which will appear multiple times throughout this section, respectively as:

**Claim E.1.** If $\gamma_t = \eta_{t+1}/\eta_t \in (0, 1]$ for each $t \geq 1$, then

$$f_t(x_t) - f_t(z) \leq \frac{1}{\eta_t}(D_\Phi(x_t, w_{t+1}) - D_\Phi(z, w_{t+1}) + D_\Phi(z, x_t)).$$

**Claim E.2.** If $\gamma_t \in (0, 1]$ for all $t \geq 1$, then,

$$\frac{1}{\eta_t}(D_\Phi(x_t, w_{t+1}) - D_\Phi(z, w_{t+1}) + D_\Phi(z, x_t))$$
$$\leq \frac{D_\Phi(x_t, w_{t+1})}{\eta_t} + \frac{1}{\eta_t}\Big(\Big(\frac{1}{\gamma_t} - 1\Big)D_\Phi(z, x_1) - \frac{1}{\gamma_t}D_\Phi(z, x_{t+1}) + D_\Phi(z, x_t)\Big).$$

## E.1  Sublinear Regret for Relative Lipschitz Functions

In this subsection we prove sublinear regret for DS-OMD with relative Lipschitz continuous cost functions. First we use Theorem 4.1 in Fang et al. [2020]. This theorem is analogous to the bound given in the analysis of classic OMD given by Bubeck [2015, Theorem 4.2].

**Theorem E.3** (Fang et al. [2020, Theorem 4.1]). If $\gamma_t := \eta_{t+1}/\eta_t$ for each $t \geq 1$, then

$$\mathrm{Regret}_T(z) \leq \sum_{t=1}^{T}\frac{D_\Phi(x_t, w_{t+1})}{\eta_t} + \frac{D_\Phi(z, x_1)}{\eta_{T+1}}, \qquad \forall T > 0.$$

Now we are ready to use Theorem E.3 to prove Theorem 5.1.

*Proof of Theorem 5.1.* We first need to bound the terms $D_\Phi(x_t, w_{t+1})$ for each $t \geq 1$. Fix $t \geq 1$. By the three-point identity for Bregman divergences (see (2.2)),

$$D_\Phi(x_t, w_{t+1}) = -D_\Phi(w_{t+1}, x_t) + \langle\nabla\Phi(x_t) - \nabla\Phi(w_{t+1}), x_t - w_{t+1}\rangle. \tag{E.1}$$

From the definition of the iterates in Algorithm 2, we have $\eta_t g_t = \nabla\Phi(x_t) - \nabla\Phi(w_{t+1})$. Thus,

$$(E.1) = -D_\Phi(w_{t+1}, x_t) + \eta_t \langle g_t, x_t - w_{t+1} \rangle,$$

$$\overset{(2.3)}{\leq} -D_\Phi(w_{t+1}, x_t) + \eta_t L\sqrt{2D_\Phi(w_{t+1}, x_t)} \leq \frac{\eta_t^2 L^2}{2}, \tag{E.2}$$

where first inequality is from (2.3) ( since $f_t$ is Lipschitz continuous relative to $\Phi$) and the second inequality comes from the fact that $\sqrt{\alpha\beta} \leq (\alpha + \beta)/2$ with $\alpha := \eta_t^2 L^2$ and $\beta := D_\Phi(w_{t+1}, x_t)$. Plugging the above in Theorem E.3, we get

$$\text{Regret}_T(z) \leq \sum_{t=1}^{T} \frac{\eta_t L^2}{2} + \frac{D_\Phi(z, x_1)}{\eta_{T+1}} \leq \sum_{t=1}^{T} \frac{\eta_t L^2}{2} + \frac{K}{\eta_{T+1}}.$$

Setting $\eta_t := \sqrt{K}/L\sqrt{t}$ for each $t \geq 1$ and by using Lemma B.1 from Appendix B we have

$$\text{Regret}_T(z) \leq \frac{L^2}{2} \cdot \frac{\sqrt{K}2\sqrt{T}}{L} + K\frac{L\sqrt{T+1}}{\sqrt{K}} \leq 2L\sqrt{K(T+1)}. \qquad \square$$

## E.2  Proof for Theorem 5.3

In this section we give a logarithmic regret bound for OMD the cost functions are when relative Lipschitz continuous and relative strongly convex, both relative to the mirror map. The first step in the proof is the following claim given by modifying Claims E.1 and E.2 and combining them together.

**Claim E.4.** Assume that $\gamma_t = 1$ for all $t \geq 1$, then

$$f_t(x_t) - f_t(z) \leq \frac{1}{\eta_t}\big(D_\Phi(x_t, w_{t+1}) - D_\Phi(z, x_{t+1}) + D_\Phi(z, x_t)\big) - MD_\Phi(z, x_t).$$

*Proof of Claim E.4.* This proof largely follows the structure of the proof of Claim E.1. First, instead of using subgradient inequality, we use the definition of relative strong convexity and get

$$f_t(x_t) - f_t(z) \leq \langle g_t, x_t - z \rangle - MD_\Phi(z, x_t).$$

By proceeding as in the proof of Claim E.1 but adding the extra term $-MD_\Phi(z, x_t)$ term we get

$$f_t(x_t) - f_t(z) \leq \frac{1}{\eta_t}\big(D_\Phi(x_t, w_{t+1}) - D_\Phi(z, w_{t+1}) + D_\Phi(z, x_t)\big) - MD_\Phi(z, x_t).$$

Then we apply Claim E.2 with $\gamma_t = 1$ to get the desired inequality. $\qquad \square$

The next step in the proof of the logarithmic regret bound is to sum Claim E.4 over $t$, yielding

$$\sum_{t=1}^{T} \big(f_t(x_t) - f_t(z)\big)$$

$$\leq \sum_{t=1}^{T} \frac{D_\Phi(x_t, w_{t+1})}{\eta_t} + \sum_{t=2}^{T}\left(\left(\frac{1}{\eta_t} - \frac{1}{\eta_{t-1}}\right)D_\Phi(z, x_t) - MD_\Phi(z, x_t)\right)$$

$$+ \frac{1}{\eta_1}D_\Phi(z, x_1) - \frac{1}{\eta_T}D_\Phi(z, x_{T+1}) - MD_\Phi(z, x_1), \qquad \text{(by Claim E.4)}$$

$$\leq \sum_{t=1}^{T} \frac{D_\Phi(x_t, w_{t+1})}{\eta_t} + \sum_{t=2}^{T}\left(\left(\frac{1}{\eta_t} - \frac{1}{\eta_{t-1}}\right)D_\Phi(z, x_t) - MD_\Phi(z, x_t)\right). \qquad (\eta_1 = 1/M)$$

Since $\eta_t = \frac{1}{Mt}$, we have

$$\sum_{t=2}^{T}\left(\left(\frac{1}{\eta_t} - \frac{1}{\eta_{t-1}}\right)D_\Phi(z, x_t) - MD_\Phi(z, x_t)\right) = \sum_{i=2}^{T}\left(MD_\Phi(z, x_t) - MD_\Phi(z, x_t)\right) = 0.$$

We have already shown that $D_\Phi(x_t, w_{t+1}) \le \frac{\eta_t^2 L^2}{2}$ in (E.2), so

$$\text{Regret}_T(z) \le \sum_{t=1}^T \frac{D_\Phi(x_t, w_{t+1})}{\eta_t} + \sum_{i=2}^T \left( \left( \frac{1}{\eta_t} - \frac{1}{\eta_{t-1}} \right) D_\Phi(z, x_t) - M D_\Phi(z, x_t) \right),$$

$$\le \sum_{t=1}^T \frac{\eta_t L^2}{2} = \frac{L^2}{2M} \sum_{t=1}^T \frac{1}{t} \le \frac{L^2}{2M} (\log T + 1).$$

The last step comes from upper bound of the harmonic series.

### E.3 Sublinear Regret for DS-OMD with Extra Regularization

Following the notation from Appendix D, we let $\Psi \colon \mathcal{X} \to \mathbb{R}_+$ denote the extra regularizer, a nonnegative convex function. We also assume $\Psi$ is minimized at $x_1$ with value $0$ and use composite regret to measure the performance. The only modification we need to make to Algorithm 2 is to change the projection step of the algorithm to

$$x_{t+1} = \operatorname*{arg\,min}_{x \in \mathbb{R}^n} \big( D_\Phi\big(x, y_{t+1}\big) + \eta_{t+1} \Psi(x) \big). \tag{E.3}$$

Here we minimize over $\mathbb{R}^n$ instead of over $\mathcal{X}$ since we can introduce the constraint of the points lying in $\mathcal{X}$ by adding to $\Psi$ the indicator function of $\mathcal{X}$. That is, by adding to $\Psi$ the function

$$\delta_{\mathcal{X}}(x) := \begin{cases} 0 & \text{if } x \in \mathcal{X}, \\ +\infty & \text{otherwise,} \end{cases} \qquad \forall x \in \mathbb{R}^n.$$

In the remainder of this section we denote by $\Pi^\Phi_{\eta_{t+1}\Psi}(y_{t+1})$ the point computed by the right-hand side of (E.3). If we pick this projection coefficient $\alpha_t$ carefully, we can get $O(\sqrt{T})$ regret, as specified by the next theorem.

**Theorem E.5.** Let $\{x_t\}_{t\ge 1}$ be given as in Algorithm 2 with composite updates and with parameters $\gamma_t := \eta_{t+1}/\eta_t$ for each $t \ge 1$. Assume that $\Psi(x_1) = 0$ and that $f_t$ is $L$-Lipschitz continuous relative to $\Phi$ for all $t \ge 1$. Let $z \in \mathcal{X}$ and $K \in \mathbb{R}$ be such that $K \ge D_\Phi(z, x_1)$. Then,

$$\text{Regret}_T^\Psi(z) \le \sum_{t=1}^T \frac{\eta_t L^2}{2} + \frac{K}{\eta_{T+1}}, \qquad \forall z \in \mathcal{X}, \forall T > 0.$$

In particular, for $\eta_t := \sqrt{K}/L\sqrt{t}$ for each $t \ge 1$, then $\text{Regret}_T^\Psi(z) \le 2L\sqrt{K(T+1)}$.

The analysis hinges on the following generalization of [Bubeck, 2015, Lemma 4.1], which can be thought as a "pythagorean Theorem" for Bregman projections.

**Lemma E.6.** Let $x \in \mathbb{R}^n$, $y \in \mathcal{D}^\circ$, and set $\bar{y} := \Pi^\Phi_{\alpha_t \Psi}(y)$. If $\bar{y} \in \mathcal{D}^\circ$, then

$$D_\Phi(x, \bar{y}) + D_\Phi(\bar{y}, y) \le D_\Phi(x, y) + \alpha_t(\Psi(x) - \Psi(\bar{y})).$$

*Proof of Lemma E.6.* By the optimality conditions of the projection, we have $\nabla \Phi(y) - \nabla \Phi(\bar{y}) \in \partial(\alpha_t \Psi)(\bar{y})$. Using the three-point identity of Bregman divergences (see (2.2)) and the subgradient inequality, we get

$$D_\Phi(x, \bar{y}) + D_\Phi(\bar{y}, y) - D_\Phi(x, y) = \langle \nabla \Phi(y) - \nabla \Phi(\bar{y}), x - \bar{y} \rangle \le \alpha_t(\Psi(x) - \Psi(\bar{y})).$$

Rearranging yields the desired inequality. $\qquad \square$

We are now ready to prove Theorem E.5.

*Proof of Theorem E.5.* To prove the theorem, we just need to show that Theorem E.3 still holds (with respect to the composite regret) in the algorithm with composite projections. We modify Claims E.1 and E.2 to get the following claim.

**Claim E.7.**

$$f_t(x_t) - f_t(z)$$
$$\le \frac{D_\Phi(x_t, w_{t+1})}{\eta_t} + \left( \frac{1}{\eta_{t+1}} - \frac{1}{\eta_t} \right) D_\Phi(z, x_1) + \frac{D_\Phi(z, x_t)}{\eta_t} - \frac{D_\Phi(z, x_{t+1})}{\eta_{t+1}} + (\Psi(z) - \Psi(x_{t+1})).$$

*Proof of Claim E.7.* Claim E.1 gives us the following inequality:

$$f_t(x_t) - f_t(z) \leq \frac{1}{\eta_t}(D_\Phi(x_t, w_{t+1}) - D_\Phi(z, w_{t+1}) + D_\Phi(z, x_t)).$$

Then we just need to modify Claim E.2 to bound the right side of the above inequality. Using Lemma E.6, we have

$$D_\Phi(z, y_{t+1}) - D_\Phi(x_{t+1}, y_{t+1}) \geq D_\Phi(z, x_{t+1}) + \alpha_t(\Psi(x_{t+1}) - \Psi(z)).$$

Then we substitute the step $D_\Phi(z, y_{t+1}) - D_\Phi(x_{t+1}, y_{t+1}) \geq D_\Phi(z, x_{t+1})$ in the original proof of Claim E.2 in Fang et al. [2020] with the above inequality plus the extra regularization term and Claim E.7 follows. □

Now the regret is bounded by

$$
\begin{aligned}
\text{Regret}_T^\Psi(z) \\
&= \sum_{t=1}^T \left( f_t(x_t) + \Psi(x_t) - f_t(z) - \Psi(z) \right), \\
&= \sum_{t=1}^T \left( \left( f_t(x_t) - f_t(z) \right) + \left( \Psi(x_t) - \Psi(z) \right) \right), \\
&\leq \sum_{t=1}^T \frac{D_\Phi(x_t, w_{t+1})}{\eta_t} + \sup_{z \in \mathcal{X}} \frac{D_\Phi(z, x_1)}{\eta_{T+1}} + \sum_{t=1}^T (\Psi(x_t) - \Psi(x_{t+1})), \\
&= \sum_{t=1}^T \frac{D_\Phi(x_t, w_{t+1})}{\eta_t} + \sup_{z \in \mathcal{X}} \frac{D_\Phi(z, x_1)}{\eta_{T+1}} + \Psi(x_1) - \Psi(x_{T+1}), \\
&\leq \sum_{t=1}^T \frac{D_\Phi(x_t, w_{t+1})}{\eta_t} + \sup_{z \in \mathcal{X}} \frac{D_\Phi(z, x_1)}{\eta_{T+1}}.
\end{aligned}
$$

The first inequality follows Claim E.7 and the last step comes from the assumption that $x_1$ is the minimizer of $\Psi$. This shows Theorem E.3 holds as desired and then the proof of Theorem E.5 follows as in Appendix E.1. □

Similarly, by setting all $f_t$ to a fixed function $f$ and taking average we get the following corollary.

**Corollary E.8.** Consider a convex function $f$ and let $x^*$ be a minimizer of $f$. Let $\Phi$ be a differentiable strictly convex mirror map such that $\mathcal{X} \subseteq \mathcal{D}^\circ$. Assume that $f$ is $L$-Lipschitz continuous to $\Phi$ and there exists non-negative $K$ such that $K \geq D_\Phi(x^*, x_1)$. Let $\{\eta_t\}_{t \geq 1}$ be a sequence of step sizes. If we pick step size $\eta_t = \frac{1}{\sqrt{t}}$, $\alpha_t = \eta_{t+1}$ and stabilization coefficient $\gamma_t = \eta_{t+1}/\eta_t$, then we have convergence rate

$$(f + \Psi)\left( \frac{1}{T} \sum_{t=1}^T x_t \right) - (f + \Psi)(x^*) \leq \frac{2L\sqrt{2K}}{\sqrt{T}}.$$