[Reviews · NeurIPS 2020]

Review 1

Summary and Contributions: The traditional online convex optimization relies on the assumption that the loss function is Lipchitz continuous. Recently, relative-Lipschitz conditions were proposed in convex optimization literature, which extends the scope of the problems one can solve using first-order methods. The submission studies a variety of online learning algorithms under the recently proposed relative-Lipschitz conditions, including follow the regularized leader, online mirror descent, regularized dual averaging, and extend the standard regret bounds to the relative setting, including both O(sqrt{T}) regret and O(log T) regret. I see some merits of the submission, but there are some major concerns that need to be addressed (see the weakness 1 below).

Strengths: 1. This is the first work showing that online mirror descent can achieve O(log T) regret for non-Euclidean distance, but of course with a stronger assumption, the relative strong convexity. 2. The regret bounds presented in Section 3 and Section 5 are simple and elegant. They look like the ``right’’ regret bound for these settings. 3. The contribution to O(sort{T}) regret under relative continuity is interesting, but it may not be the first result under similar condition (see weakness 1).

Weaknesses: 1. There is ICLR paper "Online and stochastic optimization beyond Lipschitz continuity: A Riemannian approach". Since it is closely related to the submission (both talk about online learning without Lipschitz continuity), it should be cited and carefully compared with. Although the presumed conditions are different (relative continuity and Riemannian continuity), it is unclear to me what are the examples that satisfies relative continuity but not Riemannian continuity. It 2. The work does not present interesting examples where the loss is relative continuous but not Lipschitz continuous. An example with some preliminary numerical experiments can greatly help the readers better understand the usage of these algorithms. 3. The work directly extends the similar results in convex optimization, which seems to be straight-forward, and may lack theoretical depth.

Correctness: I quickly checked the proof, and they seem to be correct.

Clarity: The paper is well written.

Relation to Prior Work: See the weakness 1.

Reproducibility: Yes

Additional Feedback:


Review 2

Summary and Contributions: This paper focuses on online convex optimization (OCO) problems, where the respective losses are not Lipschitz continuous, as it is assumed typically in the literature. More precisely, the standard Lipschitz continuity assumption is generalized by a new class of losses that of relative Lipschitz continuity, which is a direct extension of Lu's (2018) relative continuity. Building on this blanket assumption for the loss functions, the optimal regret bounds are recovered for two online iterative schemes; that of FTRL (and a particular case of it namely the Dual Averaging algorithm) and Dual-Stabilized OMD. In particular, the authors show the O(\sqrt{T}) regret bound for two scenarios: for the standard online setting and the so-called composite loss function framework. Additionally, the authors show a logarithmic regret bound for the case where the loss functions are in addition relatively strongly convex, which is a Bregman based generalization of the traditional euclidean notion of strong convexity.

Strengths: The main novelty of this paper is that it tries to establish logarithmic regret for loss functions that go beyond the traditional euclidean notions of Lipschitz continuity and strong convexity. However, this result may be questionable since the existence of loss functions that simultaneously satisfy both of these generalized notions is unclear as it becomes apparent in what follows below.

Weaknesses: My concerns about this paper concentrate on two main points. First, the main class of losses that the paper introduces, that of relative Lipschitz continuity (Def. 2.1) seems very closely related to that of Riemann Lipschitz Continuity (RLC) in Antonakopoulos et. al (2020). In particular, given that the losses are (RLC) then one can recover relative Lipschitz continuity via a direct combination of convexity and Cauchy-Schwartz inequality. Moreover, conversely every relative Lipschitz continuous loss can be seen as (RLC) if one chooses the respective Riemannian metric accordingly; this becomes even more evident for the example that the paper presents, if f(x)=x^{2} for x\in R, then one can straightforwardly choose the Riemannian metric in such a manner that the respective dual norm would be \|v\|_{x,\ast}=|v|/x and (RLC) follows. That said, this weakens significantly the contributions concerning FTRL and the like, since in Antonakopoulos et. al (2019) these results are already established. On the other hand, concerning the most intriguing part that of establishing logarithmic regret for the case where the loss functions are in addition relatively strongly convex, there is no obvious way to establish any relevant examples that satisfy simultaneously relative Lipschitz continuity and relative strong convexity, besides of course the euclidean ones. More precisely, assume that we have the static standard convex minimization problem and x^* is a solution. Then, relative strong convexity (Def 2.2) implies that for some constant M>0: <\nabla f(x),x-x^*>\geq M D(x^\ast,x) whereas due to Lu's (2019) relative continuity, we have for some L>0: <\nabla f(x),x-x^*>\leq L\sqrt{2D(x^*,x)} Hence, the above inequalities demand that the Bregman "distance" between a solution and the base point x remains bounded, which fails to be true for various practical examples like Poisson Inverse Problems and/or Support Vector Machines due to the singular behaviour of the regularizer (or its gradient) near the boundary of the feasible domain. That said, in order to justify the significance of the logarithmic regret result, the authors have to provide relevant examples of interest for this particular class of loss functions. In conclusion, in the blanket assumption part 2.1, the losses domain X is assumed to be closed. This assumption, despite the fact that it is typical for the Euclidean framework, a priori excludes various interesting problems (e.g. f(x)= -logx, x>0, the KL-divergence etc). Post rebuttal: The authors provided an example of a relatively continuous/relatively strongly convex function; therefore I increased my score. However, I think that the paper needs a revision according the following points: 1.Clarify the connection with existing work, in particular the paper of Antonakopoulos et.al. 2020. 2. Highlight in a more clear way the significance of logarithmic regret obtained for relatively continuous/relative strongly convex functions by providing a more extensive presentation and applications of the said class of fucntions.

Correctness: The proofs seem correct and sound. However, the issue mentioned above concerning the relation of relative strong convexity and relative Lipschitz continuity seems unjustified.

Clarity: The paper is well-written and easy to follow.

Relation to Prior Work: To the best of my knowledge, the most related work that deals with (OCO) problems where the respective losses are not Lipschitz continuous is that of Antonakopoulos et al (2020); in the said paper, the notion that extends the standard Lipschitz is that of Riemann Lipschitz continuity (RLC). On the other hand, in this paper. the key notion is that of relative Lipschitz continuity. In their introduction, the author(s) claim that the relation between these two notions is unclear. However, as I mentioned above, this is not true. More precisely, given (RLC) one can recover relative Lipschitz continuity via a direct combination of convexity and Cauchy-Schwartz inequality. Additionally, every relatively Lipschitz continuous loss function can be seen also as (RLC) under an. appropriately chosen Riemannian metric.

Reproducibility: Yes

Additional Feedback:


Review 3

Summary and Contributions: The paper extends known regret bounds for OCO ("Online Convex Optimization") to the case where adversary functions are "relatively Lipshitz continuous" or "relatively strongly convex." Informally, these two notions are generalizations of usual of Lipshitz continuity and strong convexity, where one replaces a norm with a Bregman divergence relative to a convex function R (the traditional setting corresponds to R=1/2||.||^2). Classical results establish that if adversary functions are Lipshitz continuous or strongly convex then certain algorithms can achieve regret of O(sqrt(T)) and O(log T) respectively. This paper shows similar results hold in the relative setting (Theorems 3.2 and 3.3). The algorithms that achieve these bounds are "follow the (regularized) leader". The authors also show that the O(sqrt(T)) bound still applies in case of more tractable (online) "dual averaging" algorithms. Finally, the authors prove that "Dual-Stabilized Online Mirror Descent" (DS-OMD) also achieves O(sqrt(T)) for relatively Lipshitz functions, and that (non-stabilized) OMD achieves O(log(T)) for relatively strongly convex functions.

Strengths: The paper is rigorous and clearly written and discusses important topics for the optimization community.

Weaknesses: My only concern is whether the results presented in the paper are sufficiently novel. In particular, the authors mention in the related work section that Antonakopoulos et al. also prove O(sqrt(T)) bounds for FTRL and OMD beyond traditional Lipschitz Contintuity (in the context of "Riemann Lipschitz Contintiity"). I would have liked a more careful comparison with this work. Is there any advantage of the "relative" framework compared to the "Riemannian" one?

Correctness: The paper is technically sound.

Clarity: The paper is clear and well written.

Relation to Prior Work: As mentioned above, I think the paper is missing a careful comparison with the work of Antonakopoulos et al.

Reproducibility: Yes

Additional Feedback: I enjoyed reading the paper even though I was not very familiar with the topic. Perhaps some motivational examples could be useful. In particular, the authors mention in the Introduction that not all loss functions that appear in applications satisfy the traditional strong convexity assumptions. Maybe the authors could expand on this point and provide a concrete setting where the more general theory is necessary.

[Author Response · NeurIPS 2020]

First of all, we would like to thank all three reviewers for their feedback. It is clear that all of them engaged with the paper and managed to give truly thoughtful feedback. In what follows we try to address some of the concerns described in the reviews:

(1) **Relationship between RLC and relative Lipschitz continuity:** All the reviewers mentioned we should have discussed the relation between RLC and relative Lipschitz continuity. We agree with that and the lack of this discussion made it hard to understand the relationship between the results of our paper and the one by Antonakopoulos et al. [2020]. Reviewer 2 mentioned a possible equivalence (or at least close relationship) between the Riemannian and relative Lipschitz continuity definitions. In one direction, if a convex (differentiable) function $f$ is RLC w.r.t. a metric $g$ and a function $R$ is strongly convex w.r.t. the metric $g$, then $f$ is relative Lipschitz w.r.t. $R$. However, in the case where $f$ is *not* differentiable, it is not clear if this implication holds. Without differentiability, the definition of RLC becomes more intricate. Even in the differentiable case, relative Lipschitzn continuity is at least as general as RLC by the previous argument. For the reverse direction, it is not in general obvious how to find a proper Riemannian metric to make the function RLC given relative Lipschitz continuity. Moreover, the simplicity of our proofs allow us to easily extend the results to the case with composite functions. Thus, in the revised version of the paper we plan to better discuss the relationship between RLC and our results based on the following points:

1. The results that require only relative Lipschitz continuity are at least as general as the ones for RLC;

2. Given a function that is relative Lipschitz, it is not in general obvious how to find a metric to make it RLC;

3. This works focuses on non-differentiable cost functions, where the relationship is less obvious.

(2) **Lack of motivating examples:** The most interesting applications we currently know of relative Lipschitz continuous functions, such as SVM training and Ellipsoid intersection detection, are described in the work of Lu [2019]. The inverse Poisson problem in Antonakopoulos et al. [2020] is also an application, given the discussion in the previous point. We plan to add a more interesting discussion of applications in the revised version, and given the generality of the results we expect to see more applications in the future.

(3) **Significance of our logarithmic regret bounds:** Reviewer 2 pointed out that requiring a function to be both Lipschitz continuous and strongly convex relative to the same regularizer may be too restrictive since it implies that the Bregman divergence between an optimal solution and any point in the feasible set $\mathcal{X}$ should be bounded. As they mentioned, this does not hold when $\mathcal{X}$ is unbounded or when the Bregman divergence explodes on the boundary of $\mathcal{X}$. However, we note that the same argument holds in the Euclidean case. This implies that even classical logarithmic regret results can only be applied in the case where $\mathcal{X}$ is bounded. Hence, it is not surprising that we cannot have an unbounded feasible set in the relative case.

It is worth mentioning that even the sublinear regret bound from Antonakopoulos et al. [2020] for the inverse Poisson problem requires a bounded feasible region, since the strong convexity w.r.t. the metric used only holds in a bounded set (they show that it holds on $[0,1]^n$ and one can get strong convexity with worse constants by scaling). That is, boundedness of the feasible region is needed even for $O(\sqrt{T})$ regret guarantee in this important example. So the necessity of a bounded feasible region for logarithmic regret does not seem significantly more restrictive.

Reviewer 2 is correct that our logarithmic regret bounds cannot hold where the Bregman divergence explodes at the boundary, but the relative setting can be useful beyond such case. For example, one path of future research is to devise new algorithms for optimization problems that already have logarithmic regret bounds although with a bad dependency on the dimension. Our bounds for the relative setting are similar to the ones for the classical settings, but the Lipschitz and strong convexity constants are deeply related to the regularizer used. Thus, different regularizers might yield widely different dependency on the dimension of the problem. A classical example is the experts problem: we have $O(\sqrt{Tn})$ regret guarantee with $n$ experts when using gradient descent (squared $\ell_2$-norm as a regularizer) but a $O(\sqrt{T \log n})$ regret guarantee when using the multiplicative weights update method (negative entropy as the regularizer). It is our hope that the results for the relative setting can yield new application with better theoretical dependence on the instance's parameters, such as the dimension.

# References

K. Antonakopoulos, E. V. Belmega, and P. Mertikopoulos. Online and stochastic optimization beyond lipschitz continuity: A riemannian approach. In *8th International Conference on Learning Representations, ICLR*, 2020.

H. Lu. "Relative continuity" for non-lipschitz nonsmooth convex optimization using stochastic (or deterministic) mirror descent. *Informs Journal on Optimization*, pages 265–352, 2019.


[Meta-Review · NeurIPS 2020]

This paper treats the problem of online convex optimization without Lipschitz continuity of the loss functions. The authors consider a variant of Lipschitz continuity called "relative Lipschitz continuity": this notion is originally due to Lu (2019) and involves a Bregman divergence instead of the standard norm in comparing nearby points. In this context, the authors prove the following results: - Under only relative Lipschitz continuity: an $O(sqrt{T})$ regret bound for follow-the-regularized-leader (FTRL) and a "stabilized" variant of the online mirror descent (OMD) algorithm. - Under relative continuity and relative strong convexity: an $O(logT)$ bound for the above algorithms. These results are similar to standard bounds in the literature for Lipschitz continuous / strongly convex functions. The extension to *relative* Lipschitz continuous / strongly convex functions was welcomed by the reviewers, but two major issues were identified: 1. An earlier ICLR paper by Antonakopoulos et al. (2020) already provides $O(\sqrt{T})$ bounds for FTRL and OMD under a closely related "Riemannian Lipschitz continuity" condition. The authors do not explain the relation between these two variants of Lipschitz continuity, and this can cause significant confusion. 2. Even though the reviewers acknowledged the significance of the authors' logarithmic regret bounds, the authors did not provide in the paper an example of a function that is simultaneously relatively Lipschitz continuous and relatively strongly convex in a non-standard way. This paper generated significant discussion during the review phase and I also reached out to an external referee for an additional opinion. The authors' rebuttal addressed point (1) but point (2) was left more open; however, the authors did provide a concrete example when I contacted them about this issue. This paper represents a "risk" in the following sense: the reviewers believe that there is sufficient merit in the authors' results (I agree with this). On the other hand, addressing the points identified above requires a rewrite of certain parts of the paper without the possibility of getting a second look at it. Given that the paper is overall well written and the required changes are "additive" instead of "restructuring" (which would be more difficult), there is consensus for an "accept" recommendation subject to the following changes: 1. The authors should include a precise definition of Riemannian Lipschitz continuity, how it relates to relative Lipschitz continuity, and a statement of the results of Antonakopoulos et al. for FTRL and OMD in order to facilitate the comparison with the paper's new results. 2. The authors should give a detailed description of how non-quadratic polynomials satisfy the relative strong convexity and Lipschitz continuity assumptions (over bounded domains) in a non-standard way. The extra page available for the camera ready should be more than enough to implement the above required additions. Subject to these changes, the paper would make a fine addition to the technical program of NeurIPS 2020.